# Domain Watermark: Effective and Harmless Dataset Copyright Protection is Closed at Hand

**Junfeng Guo**[1,*]**, Yiming Li**[2,3,*]**, Lixu Wang**[4]**, Shu-Tao Xia**[5]**, Heng Huang**[1]**, Cong Liu**[6]**, Bo Li**[7,8]

[1]Department of Computer Science, University of Maryland
[2]ZJU-Hangzhou Global Scientific and Technological Innovation Center
[3]School of Cyber Science and Technology, Zhejiang University
[4]Department of Computer Science, Northwestern University
[5]Tsinghua Shenzhen International Graduate School, Tsinghua University
[6]Department of Electronic and Computer Engineering, UC Riverside
[7]Department of Computer Science, University of Illinois Urbana-Champaign
[8]Department of Computer Science, University of Chicago
`{gjf2023,heng}@umd.edu`; `li-ym@zju.edu.cn`;
`lixuwang2025@u.northwestern.edu`;
`xiast@sz.tsinghua.edu.cn`; `congl@ucr.edu`; `lbo@illinois.edu`

## Abstract

The prosperity of deep neural networks (DNNs) is largely benefited from open-source datasets, based on which users can evaluate and improve their methods. In this paper, we revisit backdoor-based dataset ownership verification (DOV), which is currently the only feasible approach to protect the copyright of open-source datasets. We reveal that these methods are fundamentally harmful given that they could introduce malicious misclassification behaviors to watermarked DNNs by the adversaries. In this paper, we design DOV from another perspective by making watermarked models (trained on the protected dataset) correctly classify some 'hard' samples that will be misclassified by the benign model. Our method is inspired by the generalization property of DNNs, where we find a *hardly-generalized domain* for the original dataset (as its *domain watermark*). It can be easily learned with the protected dataset containing modified samples. Specifically, we formulate the domain generation as a bi-level optimization and propose to optimize a set of visually-indistinguishable clean-label modified data with similar effects to domain-watermarked samples from the hardly-generalized domain to ensure watermark stealthiness. We also design a hypothesis-test-guided ownership verification via our domain watermark and provide the theoretical analyses of our method. Extensive experiments on three benchmark datasets are conducted, which verify the effectiveness of our method and its resistance to potential adaptive methods. The code for reproducing main experiments is available at https://github.com/JunfengGo/Domain-Watermark.

## 1 Introduction

Deep neural networks (DNNs) have been applied to a wide range of domains and have shown human-competitive performance. The great success of DNNs heavily relies on the availability of various open-source datasets (*e.g.*, CIFAR [1] and ImageNet [2]). With these high-quality datasets, researchers can evaluate and improve the proposed methods upon them. Currently, most of these datasets limit their usage to education or research purpose and are prohibited from commercial applications without authorization. How to protect their copyrights is of great significance [3, 4, 5, 6].

---

[*]The first two authors contributed equally to this work. Correspondence to Yiming Li.

37th Conference on Neural Information Processing Systems (NeurIPS 2023).

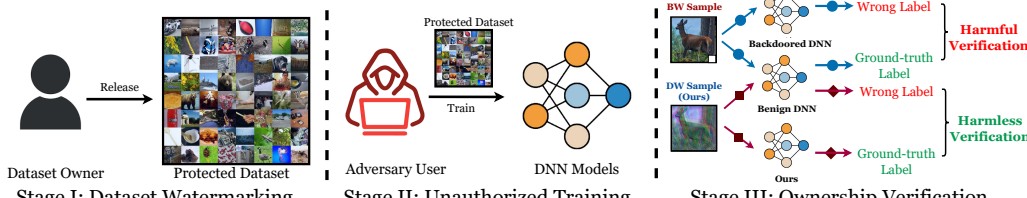

Figure 1: The main pipeline of dataset ownership verification with backdoor-based dataset watermarks and our domain watermark, where BW Sample represents existing backdoor-watermarked sample while DW Sample represents our proposed domain-watermarked sample. Existing backdoor-based methods make the watermarked model (*i.e.*, the backdoored DNN) misclassify 'easy' samples that can be correctly predicted by the benign model and therefore the verification is harmful. In contrast, our ownership verification is harmless since we make the watermarked model correctly predict 'hard' samples that are misclassified by the benign model.

Currently, there are many classical methods for data protection, such as encryption [7, 8, 9], differential privacy [10, 11, 12], and digital watermarking [13, 14, 15, 16]. However, they are not able to protect the copyrights of open-source datasets since they either hinder the dataset accessibility (*e.g.*, encryption) or require the information of the training process (*e.g.*, differential privacy and digital watermarking) of suspicious models that could be trained on it.

To the best of our knowledge, backdoor-based dataset ownership verification (DOV) [3, 4, 5] is currently the only feasible approach to protect them, where defenders exploit backdoor attacks [17, 18, 19] to watermark the original dataset such that they can verify whether a suspicious model is trained on the protected dataset by examining whether it has specific backdoor behaviors. Recently, Li *et al.* [4] first discussed the 'harmlessness' requirement of backdoor-based DOV that the dataset watermark should not introduce new security risks to models trained on the protected dataset and proposed untargeted backdoor watermarks towards harmless verification by making the predictions of watermarked samples dispersible instead of deterministic (as a pre-defined target label).

In this paper, we revisit dataset ownership verification. We argue that backdoor-based dataset watermarks can never achieve truly harmless verification since their fundamental mechanism is making watermarked model misclassifies 'easy' samples (*i.e.*, backdoor-poisoned samples) that can be correctly predicted by the benign model (as shown in Figure 1). It is with these particular misclassification behaviors that the dataset owners can conduct ownership verification. An intriguing question arises: *Is harmless dataset ownership certification possible to achieve*?

The answer to the aforementioned problem is positive. In this paper, we design dataset ownership verification from another angle, by making the watermarked model can correctly classify some 'hard' samples that will be misclassified by the benign model. Accordingly, we can exploit this difference to design ownership verification while not introducing any malicious prediction behaviors to watermarked models that will be deployed by dataset users (as shown in Figure 1). In general, our method is inspired by the generalization property of DNNs, where we intend to find a *hardly-generalized domain* for the original dataset. It can be easily learned with the protected dataset containing modified samples. Specifically, we formulate the domain generation as a bi-level optimization and leverage a transformation module to generate domain-watermarked samples; We propose to optimize a set of visually-indistinguishable modified data having similar effects to domain-watermarked samples as our *domain watermark* to ensure the stealthiness of dataset watermarking; We design a hypothesis-test-guided method to conduct ownership verification via our domain watermark at the end. We also provide theoretical analyses of all stages in our method.

In conclusion, the main contributions of this paper are four-folds: **1)** We revisit dataset ownership verification (DOV) and reveal the harmful drawback of methods based on backdoor attacks. **2)** We explore the DOV problem from another perspective, based on which we design a truly harmless DOV method via domain watermark. To the best of our knowledge, this is the first non-backdoor-based DOV method. Our work makes dataset ownership verification become an independent research field instead of the sub-field of backdoor attacks. **3)** We discuss how to design the domain watermark and provide its theoretical foundations. **4)** We conduct experiments on benchmark datasets, verifying the effectiveness of our method and its resistance to potential adaptive methods.

## 2 Related Work

### 2.1 Backdoor Attacks

Backdoor attack[2] [23, 24, 25] is a training-phrase threat of DNNs, where the adversary intends to implant a *backdoor* (*i.e.*, the latent connection between the adversary-specified trigger pattern and the target label) into the victim model by maliciously manipulating a few training samples. The backdoored DNNs behave normally while their predictions will be maliciously changed to the target label whenever the testing samples contain the trigger pattern. In general, existing backdoor attacks can be divided into two main categories based on the property of the target label, as follows:

**Poisoned-Label Backdoor Attacks.** In these attacks, the target label of poisoned samples is different from their ground-truth labels. This is the most classical attack paradigm and is more easily to implant hidden backdoors. For example, BadNets [17] is the first backdoor attack, where the adversaries randomly modify a few samples from the original dataset by attaching a pre-defined trigger patch to their images and changing their labels to the target label. These modified samples (dubbed *poisoned samples*) associated with remaining benign samples are packed as the *poisoned dataset* that is released to victim users for training; After that, Chen *et al*. [26] improved the stealthiness of BadNets by introducing trigger transparency; Nguyen *et al*. [27] proposed a more stealthy backdoor attack whose trigger patterns were designed via image-warping; Recently, Li *et al*. [4] proposed the first untargeted (poisoned-label) backdoor attack (*i.e.*, UBW-P) for dataset ownership verification.

**Clean-Label Backdoor Attacks.** In these attacks, the target label of poisoned samples is consistent with their ground-truth labels. Accordingly, these attacks are more stealthy, compared to the poisoned-label ones. However, they usually suffer from low effectiveness, especially on datasets with a high image resolution or many classes, due to the *antagonistic effects* of 'robust features' related to the target class contained in poisoned samples [28]. Label-consistent attack is the first clean-label attack where the adversaries introduced untargeted adversarial perturbations before adding trigger patterns; After that, a more effective attack (*i.e.*, Sleeper Agent [29]) is proposed, which crafts clean-label poisoned samples via bi-level optimization; Recently, Li *et al*. [4] proposed UBW-C, which generated poisoned samples for leading untargeted misclassifications to attacked DNNs.

### 2.2 Data Protection

**Classical Data Protection.** Data protection is a classical and important research direction, aiming to prevent unauthorized data usage or protect data privacy. Currently, existing classical data protection can be roughly divided into three main categories, including **(1)** encryption, **(2)** digital watermarking, and **(3)** privacy protection. Specifically, encryption [30, 7, 8] encrypts the whole or parts of the protected data so that only authorized users who hold a secret key for decryption can use it; Digital watermarking [31, 32, 33] embeds an owner-specified pattern to the protected data to claim the ownership; Privacy protection focuses on preventing the leakage of sensitive information of the data in both empirical [34, 35, 36] and certified manners [10, 37, 12]. However, these traditional approaches are not feasible to protect the copyright of open-source datasets since they either hinder the dataset accessibility or require the information of the training process that will not be disclosed.

**Dataset Ownership Verification.** Dataset ownership verification (DOV) is an emerging topic in data protection, aiming to verify whether a given suspicious model is trained on the protected dataset. To the best of our knowledge, this is currently the only feasible method to protect the copyright of open-source datasets. Specifically, it intends to implant specific prediction (towards verification samples) behaviors in models trained on the protected dataset while not reducing their performance on benign samples. Dataset owners can conduct ownership verification by examining whether the suspicious model has these behaviors. Currently, all DOV methods [3, 4, 5] exploit backdoor attacks to watermark the unprotected benign dataset. For example, [3] adopted poisoned-label backdoor attacks while [5] adopted clean-label ones for dataset watermarking. Recently, Li *et al*. [4] first discussed the 'harmlessness' requirement of DOV that the dataset watermark should not introduce new security risks to models trained on the protected dataset and proposed the untargeted backdoor watermarks. However, there is still no definition of harmlessness and backdoor-based DOV methods

---

[2] In this paper, we focus on poison-only backdoor attacks, where the adversaries can only modify a few training samples to implant backdoors. Only these attacks can be used as the dataset watermark for ownership verification. Attacks with more requirements (*e.g.*, control model training) [20, 21, 22] are out of our scope.

can never achieve truly harmless verification for they introduce backdoor threats. How to design a harmless DOV method is still an important open question.

# 3 Domain Watermark

## 3.1 Preliminaries

**Threat Model.** Following existing works in dataset ownership verification [3, 4, 5], we assume that the defenders (*i.e.*, dataset owners) can only watermark the *benign dataset* to generate the *protected dataset*. They will release the protected dataset instead of the original benign dataset for copyright protection. Given a third-party suspicious model that may be trained on the protected dataset without authorization, we consider the *black-box setting* where defenders have no information about other training configurations (*e.g.*, loss function and model architecture) of the model and can only access it to obtain predicted probability vectors via its model API.

**The Main Pipeline of Dataset Watermark.** Let $\mathcal{D} = \{(\boldsymbol{x}_i, y_i)\}_{i=1}^N$ denotes the benign training dataset. Let we consider an image classification task with $K$-classes, *i.e.*, $\boldsymbol{x}_i \in \mathcal{X} = [0, 1]^{C \times W \times H}$ represents the image with $y_i \in \mathcal{Y} = \{1, \cdots, K\}$ as its label. Instead of releasing $\mathcal{D}$ directly, the dataset owner will generate and release its watermarked version (*i.e.*, $\mathcal{D}_w$). Specifically, $\mathcal{D}_w = \mathcal{D}_m \cup \mathcal{D}_b$, where $\mathcal{D}_m$ consists of the modified version of samples from a small selected subset $\mathcal{D}_s$ of $\mathcal{D}$ (*i.e.*, $\mathcal{D}_s \subset \mathcal{D}$) and $\mathcal{D}_b$ contains remaining benign samples (*i.e.*, $\mathcal{D}_b = \mathcal{D} - \mathcal{D}_s$). The $\mathcal{D}_m$ is generated by the defender-specified image generator $G_x : \mathcal{X} \to \mathcal{X}$ and the label generator $G_y : \mathcal{Y} \to \mathcal{Y}$, *i.e.*, $\mathcal{D}_m = \{(G_x(\boldsymbol{x}), G_y(y)) | (\boldsymbol{x}, y) \in \mathcal{D}_s\}$. For example, $G_x = (\mathbf{1} - \boldsymbol{m}) \odot \boldsymbol{t} + \boldsymbol{m} \odot \boldsymbol{x}$ and $G_y = y_t$ in BadNets [17], where $\boldsymbol{m} \in \{0, 1\}^{C \times W \times H}$ is the trigger mask, $\boldsymbol{t} \in [0, 1]^{C \times W \times H}$ is the trigger pattern, $\odot$ denotes the element-wise product, and $y_t$ is the target label. In particular, $\gamma \triangleq \frac{|\mathcal{D}_m|}{|\mathcal{D}_w|}$ is called the *watermarking rate*. All models trained on the protected dataset $\mathcal{D}_w$ will have special prediction behaviors on $G_x(\boldsymbol{x})$ for ownership verification. Specifically, let $C : \mathcal{X} \to \mathcal{Y}$ denotes a third-party suspicious model that could be trained on the protected dataset, existing backdoor-based methods will examine whether it conduct unauthorized training by testing whether $C(G_x(\boldsymbol{x})) = y_t$. Since $y_t \neq y$ in most cases, these backdoor-based watermarks are harmful.

## 3.2 Problem Formulation

As described in previous sections, existing backdoor-inspired dataset ownership verification (DOV) methods [3, 4, 5] would cause malicious misclassification on watermarked samples to all models trained on the protected dataset, therefore they are harmful. *This limitation of backdoor-based DOV methods cannot be eliminated* because their inherent mechanism is to lead the watermarked model to have particular mispredicted behaviors for verification, although the misclassification could be random and less harmful [18]. In this paper, we intend to *design a truly harmless DOV method so that the watermarked models will correctly classify watermarked samples*. Before we formally define the studied problem, we first provide the definition of harmful degree of a DOV method.

**Definition 1** (Harmful and Relatively Harmful Degree). *Let $\hat{\mathcal{D}} = \{(\hat{\boldsymbol{x}}_i, y_i)\}_{i=1}^N$ indicates a set of watermarked samples used for ownership verification of a DOV method, where $\hat{\boldsymbol{x}}_i$ is the verification sample with $y_i \in \mathcal{Y}$ as its ground-truth label (instead of its given label). Let $\hat{C}$ and $C$ represent a classifier trained on the protected and unprotected datasets, respectively. The harmful degree is $H \triangleq \frac{1}{N} \sum_{i=1}^N \mathbb{I}\{\hat{C}(\hat{\boldsymbol{x}}_i) \neq y_i\}$ and the relatively harmful degree is $\hat{H} \triangleq \frac{1}{N} \left( \sum_{i=1}^N \mathbb{I}\{\hat{C}(\hat{\boldsymbol{x}}_i) \neq y_i\} - \sum_{i=1}^N \mathbb{I}\{C(\hat{\boldsymbol{x}}_i) \neq y_i\} \right)$ where $\mathbb{I}\{\cdot\}$ is the indicator function.*

To design a harmless DOV method, we intend to make watermarked DNNs correctly classify some 'hard' samples that will be misclassified by the model trained on the unprotected benign dataset. Inspired by the generalization property of DNNs, we intend to find a *hardly-generalized domain* of the benign dataset, which can be easily learned with the protected dataset containing the modified samples. In this paper, we call this watermarking method as *domain watermark*, defined as follows.

**Definition 2** (Domain Watermark). *Given a benign dataset $\mathcal{D} = \{(\boldsymbol{x}_i, y_i)\}_{i=1}^N$, let $C : \mathcal{X} \to \mathcal{Y}$ denotes a model trained on $\mathcal{D}$. Assume that $G_d$ denotes a domain generator such that $G_d(\boldsymbol{x}_i)$ owns the same ground-truth label as $\boldsymbol{x}_i$ but belongs to a hardly-generalized domain, i.e., $\sum_{(\boldsymbol{x}_i, y_i) \in \mathcal{D}} \mathbb{I}\{C(\boldsymbol{x}_i) = y_i\} \gg \sum_{(\boldsymbol{x}_i, y_i) \in \mathcal{D}} \mathbb{I}\{C(G_d(\boldsymbol{x}_i)) = y_i\}$. We intend to find a watermarked*

version of $\mathcal{D}$ (i.e., $\mathcal{D}_d$) with watermarking rate $\gamma$, such that the watermarked model $\hat{C}$ trained on it have two properties: (1) $\frac{1}{N}\sum_{(\boldsymbol{x_i},y_i)\in\mathcal{D}}\mathbb{I}\{\hat{C}(\boldsymbol{x_i})=y_i\}\geq\beta$ and (2) $\frac{1}{N}\sum_{(\boldsymbol{x_i},y_i)\in\mathcal{D}}(\mathbb{I}\{\hat{C}(\boldsymbol{x_i})=y_i\}-\mathbb{I}\{\hat{C}(G_d(\boldsymbol{x_i}))=y_i\})\leq\tau$, where $\beta,\tau\in[0,1]$ are given parameters. In this paper, $\mathcal{D}_d$ is defined as the domain watermark of the benign dataset $\mathcal{D}$.

## 3.3 Generating the Hardly-Generalized Domain

As illustrated in Definition 2, finding a hardly-generalized target domain $\mathcal{T}$ (with domain generator $G_d$) of the source domain $\mathcal{S}$ is the first step of our domain watermark. To guide the construction of the domain $\mathcal{T}$, we have the following Lemma 1.

**Lemma 1** (Generalization Bound [38]). *The bound of expected risk on a given target domain $\mathcal{T}$ is negatively associated with mutual information between features for source $\mathcal{S}$ and target $\mathcal{T}$ domains:*

$$\mathcal{R}_{\mathcal{T}}(f)\leq\mathcal{R}_{\mathcal{S}}(f)-4I(\boldsymbol{z};\hat{\boldsymbol{z}})+4H(Y)+\frac{1}{2}d_{\mathcal{H}\triangle\mathcal{H}}(p(\boldsymbol{z}),p(\hat{\boldsymbol{z}})), \tag{1}$$

*where $\mathcal{R}_{\mathcal{T}}(f)=\mathbb{E}_{(\hat{\boldsymbol{x}},y)\sim\mathcal{T}}\left[\mathbb{I}\{C(\hat{\boldsymbol{x}})\neq y\}\right]$, $\mathcal{R}_{\mathcal{S}}(f)=\mathbb{E}_{(\boldsymbol{x},y)\sim\mathcal{S}}\left[\mathbb{I}\{C(\boldsymbol{x})\neq y\}\right]$. $I(\boldsymbol{z};\hat{\boldsymbol{z}})$ is mutual information between features from $\mathcal{S}$ and $\mathcal{T}$. $d_{\mathcal{H}\triangle\mathcal{H}}(p(\boldsymbol{z}),p(\hat{\boldsymbol{z}}))$ is $\mathcal{H}\triangle\mathcal{H}$-divergence for measuring the divergence of feature marginal distributions of two domains, and $H(\cdot)$ is the entropy.*

Lemma 1 reveals the upper bound of generalization performance on $\mathcal{T}$. Since $d_{\mathcal{H}\triangle\mathcal{H}}(\cdot)$ is intractable and hard to directly optimize, as well as [38] shows that only a $I(\cdot)$ is enough for generalization across domains, *we propose to increase the expected risk on $\mathcal{T}$ by minimizing $I(\boldsymbol{z};\hat{\boldsymbol{z}})$.*

Specifically, we formulate the design of the target domain $\mathcal{T}$ (with the domain generator $G_d(\cdot;\boldsymbol{\theta})$) as a bi-level optimization, as follows:

$$\min_{\boldsymbol{\theta}}\mathbb{E}_{p(\boldsymbol{z},\hat{\boldsymbol{z}})}\left[I(\boldsymbol{z}(\boldsymbol{w^*});\hat{\boldsymbol{z}}(\boldsymbol{\theta},\boldsymbol{w^*}))+\lambda_1\mathcal{L}_c(\boldsymbol{z}(\boldsymbol{w^*}),\hat{\boldsymbol{z}}(\boldsymbol{\theta},\boldsymbol{w^*}))\right], \tag{2}$$

$$s.t.\ \boldsymbol{w^*}=\arg\min_{\boldsymbol{w}}\left[\mathbb{E}_{(\boldsymbol{x},y)\sim\mathcal{D}}\left[\mathcal{L}(f(G_d(\boldsymbol{x};\boldsymbol{\theta});\boldsymbol{w}),y)+\mathcal{L}(f(\boldsymbol{x};\boldsymbol{w}),y)\right]-\lambda_2\mathbb{E}_{p(\boldsymbol{z},\hat{\boldsymbol{z}})}[I(\boldsymbol{z}(\boldsymbol{w});\hat{\boldsymbol{z}}(\boldsymbol{w}))]\right],$$

where $\lambda_1$, $\lambda_2$ are two positive hyper-parameters, and $\mathcal{L}(\cdot)$ is the loss function (*e.g.*, cross entropy).

Following previous works [38] in domain adaption and generalization, we propose to optimize the upper bound approximation for $I(\boldsymbol{z};\hat{\boldsymbol{z}})$ instead of itself and leverage a transformation module consisting of multiple convolutional operations as $G_d(\cdot;\boldsymbol{\theta})$ to generate the domain-watermarked image $\hat{\boldsymbol{x}}$. Specifically, we aim to craft $\hat{\boldsymbol{x}}$ via minimizing the upper bound approximation for mutual information [39] between $\boldsymbol{x}\in\mathcal{D}$ and $\hat{\boldsymbol{x}}$ in the latent feature space $\mathbb{Z}$:

$$I(\boldsymbol{z};\hat{\boldsymbol{z}})=\mathbb{E}_{p(\boldsymbol{z},\hat{\boldsymbol{z}})}\left[\log\frac{p(\hat{\boldsymbol{z}}|\boldsymbol{z})}{p(\hat{\boldsymbol{z}})}\right]\leq\mathbb{E}_{p(\boldsymbol{z},\hat{\boldsymbol{z}})}[\log p(\hat{\boldsymbol{z}}|\boldsymbol{z})]-\mathbb{E}_{p(\boldsymbol{z})p(\hat{\boldsymbol{z}})}[\log p(\hat{\boldsymbol{z}}|\boldsymbol{z})], \tag{3}$$

where $\boldsymbol{z}$ and $\hat{\boldsymbol{z}}$ are the latent vectors obtained by passing $\boldsymbol{x}$ and $\hat{\boldsymbol{x}}$ through $f(\cdot;\boldsymbol{w})$'s feature extractor. $\mathcal{L}_c(\cdot)$ is the class-conditional maximum mean discrepancy (MMD) computed on the latent space $\mathbb{Z}$ and proposed to limit the potential semantic information distortion between $\boldsymbol{x}$ and $\hat{\boldsymbol{x}}$, follows:

$$\mathcal{L}_c(z,\hat{z})=\frac{1}{K}\sum_{j=1}^{K}\left(||\frac{1}{n_s^j}\sum_{i=1}^{n_s^j}\phi(\boldsymbol{z_i^j})-\frac{1}{n_t^j}\sum_{i=1}^{n_t^j}\phi(\hat{\boldsymbol{z_i^j}})||^2\right), \tag{4}$$

where $n_s^j$, $n_t^j$ represent the number for $\boldsymbol{x}$ and $\hat{\boldsymbol{x}}$ from class $j$, and $\phi(\cdot)$ is the kernel function.

The configurations, parameter selections, and model architectures are included in Appendix A.

## 3.4 Generating the Protected Dataset

Once we obtain the hard-generalized domain generator $G_d$ with the method proposed in Section 3.3, the next step is to generate the protected dataset based on it. Before we present its technical details, we first deliver some insight into the data quantity impact for the domain watermark.

**Theorem 1** (Data Quantity Impact). *Suppose in PAC Bayesian [40], for a target domain $\mathcal{T}$ and a source domain $\mathcal{S}$, any set of voters (candidate models) $\mathcal{H}$, any prior $\pi$ over $\mathcal{H}$ before any training, any $\xi \in (0,1]$, any $c > 0$, with a probability at least $1 - \xi$ over the choices of $S \sim \mathcal{S}^{n_s}$ and $T \sim \mathcal{T}_{\mathcal{X}}^{n_t}$, for the posterior $f$ over $\mathcal{H}$ after the joint training on $S$ and $T$, we have*

$$
\mathcal{R}_{\mathcal{T}}(f) \leq \frac{c}{2(1 - e^{-c})} \widehat{\mathcal{R}}_T(f) + \frac{c}{1 - e^{-c}} \beta_{\infty}(\mathcal{T}\|\mathcal{S}) \widehat{\mathcal{R}}_S(f) + \Omega
$$
$$
+ \frac{1}{1 - e^{-c}} \left( \frac{1}{n_t} + \frac{\beta_{\infty}(\mathcal{T}\|\mathcal{S})}{n_s} \right) \left( 2\mathrm{KL}(f\|\pi) + \ln \frac{2}{\xi} \right),
\tag{5}
$$

*where $\widehat{\mathcal{R}}_T(f)$ and $\widehat{\mathcal{R}}_S(f)$ are the target and source empirical risks measured over target and source datasets $T$ and $S$, respectively. $\Omega$ is a constant and $\mathrm{KL}(\cdot)$ is the Kullback–Leibler divergence. $\beta_{\infty}(\mathcal{T}\|\mathcal{S})$ is a measurement of discrepancy between $\mathcal{T}$ and $\mathcal{S}$ defined as*

$$
\beta_{\infty}(\mathcal{T}\|\mathcal{S}) = \sup_{(\boldsymbol{x},y) \in \mathrm{SUPP}(\mathcal{S})} \left( \frac{\mathcal{P}_{(\boldsymbol{x},y) \in \mathcal{T}}}{\mathcal{P}_{(\boldsymbol{x},y) \in \mathcal{S}}} \right) \geq 1,
\tag{6}
$$

*where $\mathrm{SUPP}(\mathcal{S})$ denotes the support of $\mathcal{S}$. When $\mathcal{S}$ and $\mathcal{T}$ are identical, $\beta_{\infty}(\mathcal{T}\|\mathcal{S}) = 1$.*

Theorem 1 reveals the upper bound of $\mathcal{R}_{\mathcal{T}}(f)$ is negatively associated with the number of samples for source and target domains (*i.e.*, $n_t$ and $n_s$). Assuming $n_t$ is fixed, increasing $n_s$ can still increase generalization on the target domain. As such, it is possible to combine some domain-watermarked samples with benign samples to achieve target domain generalization. Its proof is in Appendix B.

In general, the most straightforward method to generate our domain watermark for the protected dataset is to *randomly select a few samples $(\boldsymbol{x}, y)$ from the original dataset $\mathcal{D}$ and replace them with their domain-watermarked version $(G_d(\boldsymbol{x}), y)$*. However, as we will show in the experiments, the domain-watermarked image is usually significantly different from its original version. Accordingly, the adversaries may notice watermarked samples and try to remove them to bypass our defense. To ensure the stealthiness of our domain watermark, we propose to *optimize a set of visually-indistinguishable modified data $\{(\boldsymbol{x}'_i, y_i)|\boldsymbol{x}'_i = \boldsymbol{x}_i + \boldsymbol{\delta}_i\}$ having similar effects to domain-watermarked samples*. This is also a bi-level optimization problem, as follows.

$$
\min_{\boldsymbol{\delta} \subset \mathcal{B}} \left[ \mathbb{E}_{(\hat{\boldsymbol{x}},y) \sim \mathcal{T}}[\mathcal{L}\left(f(\hat{\boldsymbol{x}}; \boldsymbol{w}(\boldsymbol{\delta})), y\right)] - \lambda_3 \min \left\{ \mathbb{E}_{(\overline{\boldsymbol{x}},y) \sim \overline{\mathcal{T}}}[\mathcal{L}\left(f(\overline{\boldsymbol{x}}; \boldsymbol{w}(\boldsymbol{\delta})), y\right)], \lambda_4 \right\} \right],
\tag{7}
$$

$$
s.t.\ \boldsymbol{w}(\boldsymbol{\delta}) = \arg\min_{\boldsymbol{w}} \left[ \frac{1}{|\mathcal{D}_s|} \sum_{(\boldsymbol{x}_i, y_i) \in \mathcal{D}_s} \mathcal{L}\left(f(\boldsymbol{x}_i + \boldsymbol{\delta}_i; \boldsymbol{w}), y_i\right) + \frac{1}{|\mathcal{D}_b|} \sum_{(\boldsymbol{x_j}, y_j) \in \mathcal{D}_b} \mathcal{L}\left(f(\boldsymbol{x}_j; \boldsymbol{w}), y_j\right) \right],
$$

where $\mathbb{E}_{(\overline{\boldsymbol{x}},y) \sim \overline{\mathcal{T}}}[\mathcal{L}\left(f(\overline{\boldsymbol{x}}; \boldsymbol{w}(\boldsymbol{\delta})), y\right)]$ represents the expected risk for the watermarked model on other unseen domains (*i.e.*, $\overline{\mathcal{T}}$) and $\mathcal{B} = \{\boldsymbol{\delta} : ||\boldsymbol{\delta}||_{\infty} \leq \epsilon\}$ where $\epsilon$ is a visibility-related hyper-parameter.

The second term in Eq.(7) is to prevent the watermarked model can achieve a similar generalization performance on other unseen domains as the target domain $\mathcal{T}$ to preserve the uniqueness of $\mathcal{T}$ for verification purposes. We introduce two parameters $\lambda_3$ and $\lambda_4$ for preventing the second term dominant in the optimization procedure. $\lambda_4$ is set as $\mathbb{E}_{(\overline{\boldsymbol{x}},y) \sim \overline{\mathcal{T}}}[\mathcal{L}\left(f(\overline{\boldsymbol{x}}; \boldsymbol{w}^*), y\right)]$, where $\boldsymbol{w}^*$ is obtained by training on the original dataset $\mathcal{D}$. Please find more optimization details in Appendix C.

In particular, our domain watermark is clean-label, *i.e.*, we do not modify the label of modified samples as have done in most backdoor-based methods. As such, it is more stealthy.

## 4 Dataset Ownership Verification via Domain Watermark

In this section, we introduce how to conduct dataset ownership verification via our domain watermark. The overview of the entire procedure is shown in Figure 2.

As described in Section 3.2, models trained on our protected dataset (with domain watermark) can correctly classify some domain-watermarked samples while other benign models cannot. Accordingly, given a suspicious third-party model $f$, the defenders can verify whether it was trained on the protected dataset by examining whether the model has similar prediction behaviors on benign samples and their domain-watermarked version. *The model is regarded as trained on the protected*

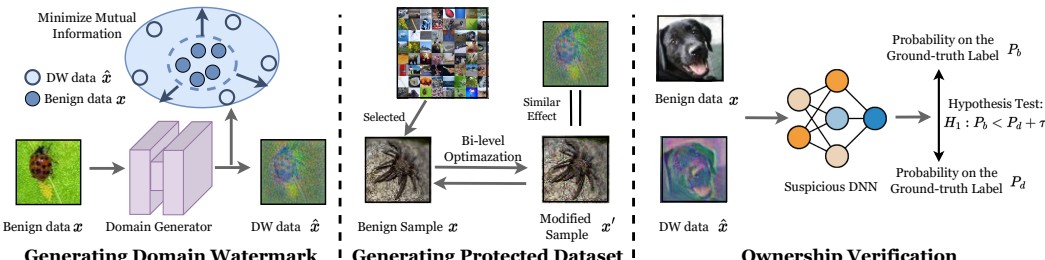

Figure 2: The workflow of dataset ownership via our domain watermark. In the first step, we will generate domain-watermarked (DW) samples in a hardly-generalized domain of the benign dataset; In the second step, we will optimize a set of visually-indistinguishable modified samples that have similar effects to domain-watermarked samples. We will release those modified samples associated with remaining benign samples instead of the original dataset for copyright protection; In the third step, we identify whether a given third-party model is trained on our protected dataset by testing whether it has similar prediction behaviors in benign images and their DW version.

*dataset if it has similar behaviors.* To verify it, we design a hypothesis-test-guided method following previous works [3, 4], as follows.

**Proposition 1.** *Suppose $f(\boldsymbol{x})$ is the posterior probability of $\boldsymbol{x}$ predicted by the suspicious model. Let variable $\boldsymbol{X}$ denotes the benign image and variable $\boldsymbol{X}'$ is its domain-watermarked version (i.e., $\boldsymbol{X}' = G_d(\boldsymbol{X})$), while variable $P_b = f(\boldsymbol{X})_Y$ and $P_d = f(\boldsymbol{X}')_Y$ indicate the predicted probability on the ground-truth label $Y$ of $\boldsymbol{X}$ and $\boldsymbol{X}'$, respectively. Given the null hypothesis $H_0 : P_b = P_d + \tau$ ($H_1 : P_b < P_d + \tau$) where the hyper-parameter $\tau \in [0, 1]$, we claim that the suspicious model is trained on the protected dataset (with $\tau$-certainty) if and only if $H_0$ is rejected.*

In practice, we randomly sample $m$ different benign samples to conduct the pairwise T-test [41] and calculate its p-value. The null hypothesis $H_0$ is rejected if the p-value is smaller than the significance level $\alpha$. Besides, we also calculate the *confidence score* $\Delta P = P_b - P_d$ to represent the verification confidence. *The smaller the $\Delta P$, the more confident the verification.*

**Theorem 2.** *Let $f(\boldsymbol{x})$ is the posterior probability of $\boldsymbol{x}$ predicted by the suspicious model, variable $\boldsymbol{X}$ denotes the benign sample with label $Y$, and variable $\boldsymbol{X}'$ is the domain-watermarked version of $\boldsymbol{X}$. Assume that $P_b \triangleq f(\boldsymbol{X})_Y > \eta$. We claim that dataset owners can reject the null hypothesis $H_0$ at the significance level $\alpha$, if the verification success rate (VSR) $V$ of $f$ satisfies that*

$$\sqrt{m-1} \cdot (V - \eta + \tau) - t_\alpha \cdot \sqrt{V - V^2} > 0, \tag{8}$$

*where $t_\alpha$ is $\alpha$-quantile of t-distribution with $(m-1)$ degrees of freedom and $m$ is sample size.*

In general, Theorem 2 indicates that our dataset verification can succeed if the VSR of the suspicious model $f$ is higher than a threshold (which is not necessarily 100%). In particular, the assumption of Theorem 2 can be easily satisfied by using benign samples that can be correctly classified with high confidence. Its proof is included in Appendix D.

## 5 Experiments

In this section, we conduct experiments on CIFAR-10 [1] and Tiny-ImageNet [42] with VGG [43] and ResNet [44], respectively. Results on STL-10 [45] are in Appendix F.

### 5.1 The Performance of Domain Watermark

**Settings.** We select seven baseline methods, containing three clean-label backdoor watermarks (*i.e.*, Label-Consistent, Sleeper Agent, and UBW-C) and four poisoned-label watermarks (*i.e.*, BadNets, Blended, WaNet, and UBW-P). Following the previous work [4], we set the watermarking rate $\gamma = 0.1$, perturbation constraint $\epsilon = 16/255$ in all cases, and adopt the same watermark patterns and parameters. The example of samples used in different watermarks is shown in Figure 3. For our

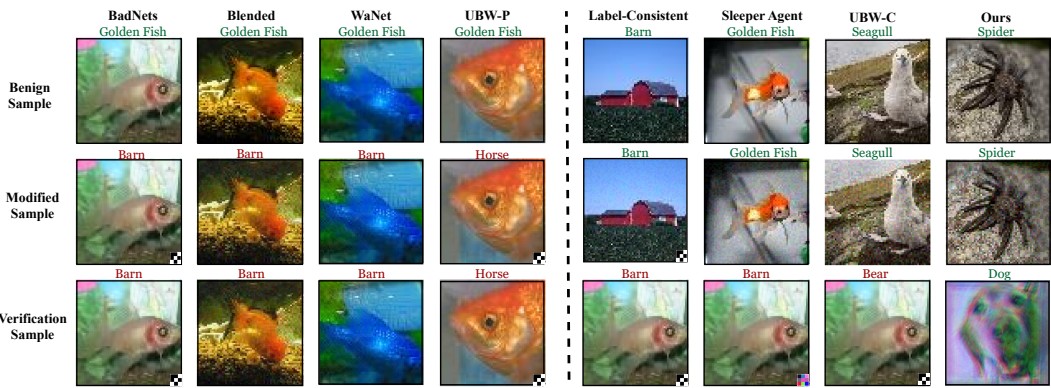

Figure 3: The example of samples involved in different methods of dataset ownership verification.

Table 1: The watermark performance on CIFAR-10 and Tiny-ImageNet datasets. In particular, we mark harmful watermark results (*i.e.*, $H > 0.5$ and $\hat{H} > 0$) in red.

| | | CIFAR-10 | | | | Tiny-ImageNet | | | |
|---|---|---|---|---|---|---|---|---|---|
| Label Type↓ | Method↓, Metric→ | BA (%) | VSR (%) | $H$ | $\hat{H}$ | BA (%) | VSR (%) | $H$ | $\hat{H}$ |
| Poisoned-Label | BadNets | 91.54 | 100 | 1.00 | 0.91 | 60.02 | 100 | 1.00 | 0.60 |
| | Blended | 91.60 | 99.96 | 1.00 | 0.92 | 59.86 | 99.97 | 1.00 | 0.60 |
| | WaNet | 90.61 | 97.3 | 0.97 | 0.87 | 57.29 | 96.14 | 0.96 | 0.51 |
| | UBW-P | 91.47 | 84.52 | 0.85 | 0.76 | 56.14 | 84.09 | 0.84 | 0.44 |
| Clean-Label | Label-Consistent | 91.64 | 99.98 | 1.00 | 0.91 | 56.92 | 41.76 | 0.61 | 0.21 |
| | Sleeper Agent | 90.73 | 93.24 | 0.93 | 0.84 | 56.82 | 87.44 | 0.87 | 0.47 |
| | UBW-C | 86.32 | 89.06 | 0.89 | 0.80 | 51.79 | 81.20 | 0.81 | 0.41 |
| | DW (Ours) | 90.86 | 90.45 | 0.10 | -0.77 | 59.10 | 58.06 | 0.42 | -0.52 |

Table 2: The effectiveness of dataset ownership verification via our domain watermark.

| | CIFAR-10 | | | Tiny-ImageNet | | |
|---|---|---|---|---|---|---|
| | Independent-D | Independent-M | Malicious | Independent-D | Independent-M | Malicious |
| $\Delta P$ | 0.79 | 0.80 | 0.04 | 0.50 | 0.67 | 0.10 |
| p-value | 1.00 | 1.00 | $10^{-54}$ | 0.90 | 1.00 | $10^{-6}$ |

method, we set $\lambda_3 = 0.3$. We implement all baseline methods based on `BackdoorBox` [46]. Each result is averaged over five runs. Please find more details in Appendix E.

**Evaluation Metrics.** We adopt benign accuracy (BA) and verification success rate (VSR) to verify the effectiveness of dataset watermarks. Specifically, the VSR is defined as the percentage that verification samples can be classified as the assigned label (*i.e.*, target label of baselines and ground-truth label of our method) by watermarked DNNs. We exploit harmful degree ($H \in [0, 1]$), and relatively harmful degree ($\hat{H} \in [-1, 1]$) to measure watermark harmfulness. In general, the larger the BA and VSR while the smaller the $H$ and $\hat{H}$, the better the dataset watermark.

**Results.** As shown in Table 1, the benign accuracy of our domain watermark is higher than clean-label backdoor watermarks in most cases, especially on the Tiny-ImageNet dataset. In particular, only our method is harmless. For example, both $H$ and $\hat{H}$ are 0.7 smaller than those of all baseline methods on CIRAR-10 datasets. Besides, as we will show in the next subsection, the VSR of our method is sufficiently high for correct ownership verification, although the VSR of our method is smaller than that of backdoor-based watermarks (especially on complicated datasets). The VSRs of benign models with our domain-watermarked samples on CIFAR-10 and Tiny-ImageNet are merely 13% and 6%, respectively. This mild potential limitation is because our VSR is restricted by the BA of watermark models. It is an unavoidable sacrifice for harmlessness.

## 5.2 The Performance of Dataset Ownership Verification via Domain Watermark

**Settings.** We evaluate our domain-watermark-based dataset ownership verification under three scenarios, including **1)** independent domain (dubbed 'Independent-D'), **2)** independent model (dubbed

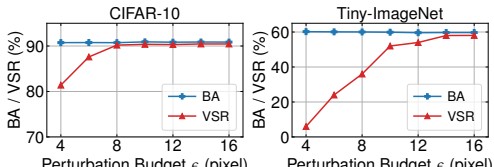

Figure 4: Effects of the perturbation budget $\epsilon$.

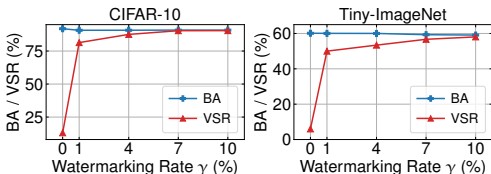

Figure 5: Effects of the watermarking rate $\gamma$.

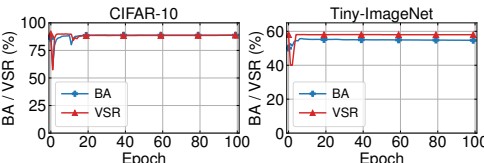

Figure 6: The resistance to fine-tuning.

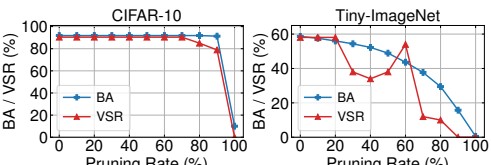

Figure 7: The resistance to model pruning.

'Independent-M'), and **3)** unauthorized dataset training (dubbed 'Malicious'). In the first case, we used domain-watermarked samples to query the suspicious model trained with modified samples from another domain; In the second case, we test the benign model with our domain-watermarked samples; In the last case, we test the domain-watermarked model with corresponding domain-watermarked samples. Notice that only the last case should be regarded as having unauthorized dataset use. More setting detail are described in Appendix G.

**Evaluation Metrics.** Following the settings in [4], we use $\Delta P \in [-1, 1]$ and $p$-value $\in [0, 1]$ for the evaluation. For the first two independent scenarios, a large $\Delta P$ and $p$-value are expected. In contrast, for the third scenario, the smaller $\Delta P$ and $p$-value, the better the verification.

**Results.** As shown in Table 2, our method can achieve accurate verification in all cases. Specifically, our approach can identify the unauthorized dataset usage with high confidence (*i.e.*, $\Delta P \approx 0$ and $p$-value $\ll 0.01$), while not misjudging when there is no unauthorized dataset utilization (*i.e.*, $\Delta P \gg 0$ and $p$-value $\gg 0.05$). Especially on the CIFAR-10 dataset (with high VSR), the $p$-values of independent cases are already 1 while that of the malicious scene is 50 powers smaller than a correct verification needs. These results verify the effectiveness of our dataset ownership verification.

## 5.3 Discussions

### 5.3.1 Ablation Studies

We hereby discuss the effects of two key hyper-parameters involved in our method (*i.e.*, $\epsilon$ and $\gamma$). Please find more experiments regarding other parameters and detailed settings in Appendix I.

**Effects of Perturbation Budget $\epsilon$.** We study its effects on both CIFAR-10 and Tiny-ImageNet datasets. As shown in Figure 4, the VSR increases with the increase of $\epsilon$. In contrast, the BA remains almost stable with different $\epsilon$. However, increasing $\epsilon$ would also reduce the invisibility of modified samples. Defenders should assign it based on their specific needs.

**Effects of watermarking Rate $\gamma$.** As shown in Figure 6, similar to the phenomena of $\epsilon$, the VSR increases with the increase of $\gamma$ while the BA remains almost unchanged on both datasets. In particular, even with a low watermarking rate (*e.g.*, 1%), our method can still have a promising VSR. These results verify the effectiveness of our domain watermark again.

### 5.3.2 The Resistance to Potential Adaptive Methods

We notice that the adversaries may try to detect or even remove our domain watermark based on existing methods in practice. In this section, we discuss whether our method is resistant to them.

Due to the limitation of space, following the previous work [4], we only evaluate the robustness of our domain watermark under fine-tuning [47] and model pruning [48] in the main manuscript. As shown in Figure 6, fine-tuning has minor effects on both the VSR and the BA of our method. Our method is also resistant to model pruning for the BA decreases with the decrease of VSR. We have

also evaluated our domain watermark to more other representative adaptive methods. Please find more setting details and results in our Appendix J.

### 5.3.3 A Closer Look to the Effectiveness of our Method

In this section, we intend to further explore the mechanisms behind the effectiveness of our domain watermark. Specifically, we adopt t-SNE [49] to visualize the feature distribution of different types of samples generated by the benign model and its domain-watermarked version. As shown in Figure 8, the domain-watermarked samples stay away (with the normalized distance as 1.84) from those with their ground-truth label (*i.e.*, '0'), although they still cluster together, under the benign model. In contrast, these domain-watermarked samples lay close (with the normalized distance as 0.40) to benign samples having the same class under the watermarked model.

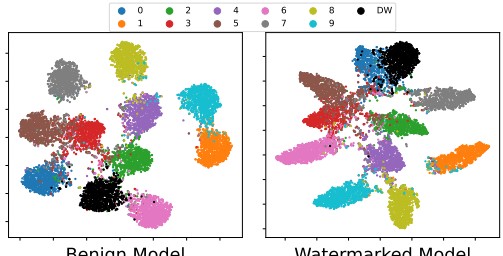

Figure 8: The t-SNE of feature representations of samples for benign and watermarked models on the CIFAR-10 dataset. The target label is '0'.

These phenomena are consistent with the predictive behaviors of the two models and can partly explain the mechanism of our domain watermark. We will further explore it in our future works.

## 6  Conclusion

In this paper, we revisited the dataset ownership verification (DOV). We revealed the harmful nature of existing backdoor-based methods because their principle is making watermarked models misclassify 'easy' samples. To design a genuinely harmless DOV method, we proposed the domain watermark by leading watermarked DNNs to correctly classify some defender-specified 'hard' samples. We provided the theoretical analyses of our domain watermark and its corresponding ownership verification. We also verified its effectiveness on benchmark datasets. As the first non-backdoor-based method, our method can provide new angles and understanding to the design of dataset ownership verification to facilitate trustworthy dataset sharing.

## Acknowledgments

Junfeng Guo and Heng Huang were partially supported by NSF IIS 1838627, 1837956, 1956002, 2211492, CNS 2213701, CCF 2217003, and DBI 2225775. Cong Liu was supported by the National Science Foundation under Grants CNS Career 2230968, CPS 2230969, CNS 2300525, CNS 2343653, and CNS 2312397.

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

**Appendix**

## Table of Contents

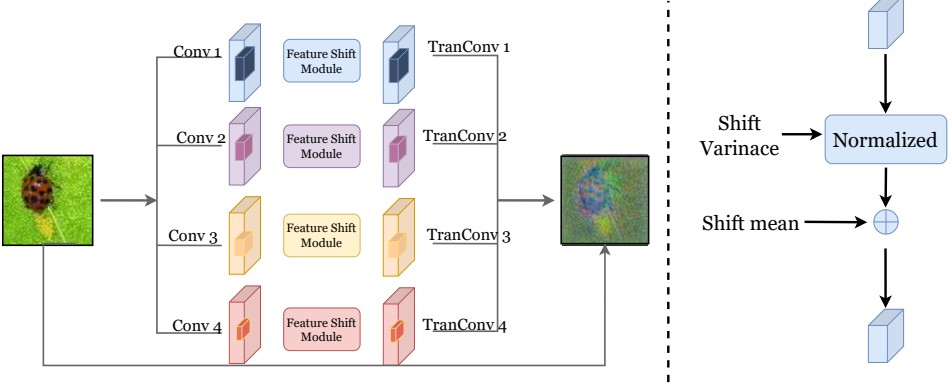

Overview of architectures for $G_d$      Details of Feature Shift Module

Figure 9: The architecture of $G_d$.

# A  Technical Details for Generating Hardly-Generalized Domain

This process is mainly motivated by [50, 51], which leveraged a transformation module with different convolution transformations to minimize the mutual information ($I$) between features from the source dataset (*i.e.*, $z$) and data from the target domain (*i.e.*, $\hat{z}$). Our work is also partially inspired by previous work on generating unlearnable samples [52], which crafted effective unlearnable samples by performing the bi-level optimization within each iteration.

## A.1  The Implementation of Transformation Module

We follow previous work [53, 38, 50] to implement the transformation module for generating samples from a different domain (as shown in Fig. 9). Specifically, we design the transformation module as an ensemble of multiple (*i.e.*, 4) convolution operations. Each convolution operation contains a convolution layer `Conv`, a feature shift module, and a corresponding transposed convolution layer `TranConv`. The detailed parameters for each convolution layer $\text{Conv}_i$ are detailed in Tab. 3. Following each convolution layer $\text{Conv}_i$, we add a feature shift module to enhance the diversity of the generated samples. Specifically, each feature shift module contains two learnable parameters $\mu_i$ and $\sigma_i$ as mean shift and variance shift, following:

$$\sigma_i \cdot \frac{\text{Conv}_i(\boldsymbol{x}) - \mu}{\sigma} + \mu_i, \tag{9}$$

where $\mu$ and $\sigma$ represent the mean and covariance value for $\text{Conv}_i(\boldsymbol{x})$. Notably, $\mu$ and $\sigma$ are not learnable parameters. Moreover, the parameters $\mu_i$ and $\sigma_i$ has the same dimension as the output of $\text{Conv}_i(\boldsymbol{x})$. After that, we use a transposed convolution layer `TranConv` to turn the feature maps generated by the above operations into a real instance, which has the same dimension as $\boldsymbol{x}$.

Putting all above, we generate the hard-generalized domain samples $\hat{\boldsymbol{x}}$ following:

$$\hat{\boldsymbol{x}} = \frac{1}{\sum w_i} \sum_i w_i \cdot \text{tahn}(\text{TranConv}(\sigma_i \cdot \frac{\text{Conv}_i(\boldsymbol{x}) - \mu}{\sigma} + \mu_i)), \tag{10}$$

where tahn represents the tahn activation function. $w_i$ is a scalar and weights the contribution of each activated instance produced by `TransposedConv` to $\hat{\boldsymbol{x}}$. $w_i$ is randomly sampled from normal distribution $w_i \sim N(0, 1)$. Notably, for each input $\boldsymbol{x}$, we first up-sample it to $224 \times 224$ size and down-sample produced $\hat{\boldsymbol{x}}$ to the original size for $\boldsymbol{x}$.

## A.2  The Optimization Process

During the optimization process of Eq. (2), we first initialized a surrogate model $f(\cdot; \boldsymbol{w})$ and a benign dataset $\mathcal{D}$. Then during each iteration for solving the bi-level optimization Eq. (2), we first minimize the $I(\boldsymbol{z}; \hat{\boldsymbol{z}})$ and $\mathcal{L}_c$ by optimizing the parameters of our proposed transformation module:

$$\min_{\boldsymbol{\theta}} \mathbb{E}_{p(\boldsymbol{z}, \hat{\boldsymbol{z}})} \left[ I(\boldsymbol{z}(\boldsymbol{w}^*); \hat{\boldsymbol{z}}(\boldsymbol{\theta}, \boldsymbol{w}^*)) + \lambda_1 \mathcal{L}_c(\boldsymbol{z}(\boldsymbol{w}^*), \hat{\boldsymbol{z}}(\boldsymbol{\theta}, \boldsymbol{w}^*)) \right]. \tag{11}$$

Table 3: The configuration for each convolution layer.

| Layer | Kernel Size | Input Channel | Output Channel |
|-------|-------------|---------------|----------------|
| $\text{Conv}_1$ | 5x5 | 3 | 3 |
| $\text{Conv}_2$ | 9x9 | 3 | 3 |
| $\text{Conv}_3$ | 13x13 | 3 | 3 |
| $\text{Conv}_4$ | 17x17 | 3 | 3 |

After that, we maximize $I(\boldsymbol{z}; \hat{\boldsymbol{z}})$ and minimize the training loss by optimizing the parameters $\boldsymbol{w}$:

$$\min_{\boldsymbol{w}} \left[ \mathbb{E}_{(\boldsymbol{x},y)\sim\mathcal{D}} \left[ \mathcal{L}(f(G_d(\boldsymbol{x};\boldsymbol{\theta}); \boldsymbol{w}), y) + \mathcal{L}(f(\boldsymbol{x};\boldsymbol{w}), y) \right] - \lambda_2 \mathbb{E}_{p(\boldsymbol{z},\hat{\boldsymbol{z}})}[I(\boldsymbol{z}(\boldsymbol{w}); \hat{\boldsymbol{z}})] \right]. \tag{12}$$

Since $I(\boldsymbol{z}; \hat{\boldsymbol{z}})$ is intractable, we propose to optimize its upper bound instead:

$$I(\boldsymbol{z}; \hat{\boldsymbol{z}}) = \mathbb{E}_{p(\boldsymbol{z},\hat{\boldsymbol{z}})} \left[ \log \frac{p(\hat{\boldsymbol{z}}|\boldsymbol{z})}{p(\hat{\boldsymbol{z}})} \right] \leq \mathbb{E}_{p(\boldsymbol{z},\hat{\boldsymbol{z}})}[\log p(\hat{\boldsymbol{z}}|\boldsymbol{z})] - \mathbb{E}_{p(\boldsymbol{z})p(\hat{\boldsymbol{z}})}[\log p(\hat{\boldsymbol{z}}|\boldsymbol{z})]. \tag{13}$$

Since the conditional distribution $p(\hat{\boldsymbol{z}}|\boldsymbol{z})$ is also intractable thus the upper bound of $I(\boldsymbol{z}; \hat{\boldsymbol{z}})$ can't be optimized, we follow previous work to adopt a variational distribution $q(\hat{\boldsymbol{z}}|\boldsymbol{z})$ to approximate the upper bound of $I(\boldsymbol{z}; \hat{\boldsymbol{z}})$:

$$I(\boldsymbol{z}; \hat{\boldsymbol{z}}) \leq \frac{1}{N} \sum_{i=1}^{N} [\log q(\hat{\boldsymbol{z}_i}|\boldsymbol{z_i}) - \frac{1}{N} \sum_{j=1}^{N} \log q(\hat{\boldsymbol{z}_j}|\boldsymbol{z_i})], \tag{14}$$

where $q(\hat{\boldsymbol{z}}|\boldsymbol{z})$ is obtained by employing the backbone neural network to approximate.

We optimize the above bi-level optimization Eq. (2) with 100 iterations. We set the learning rate as 0.005 for optimizing the parameters of the proposed transformation module and 0.001 for parameters for the backbone model $f(\cdot)$ following [50]. The batch size is 64. For both the transformation module and the backbone model $f(\cdot)$, we use SGD [54] as the optimizer with Nesterov momentum and weight decay rate of 0.0005. We use ResNet-18 as the backbone model for extracting $\boldsymbol{z}$ and $\hat{\boldsymbol{z}}$ throughout the paper. We introduce $\lambda_1$ and $\lambda_2$ for balancing each optimization objective. Following the implementation of [50], we set $\lambda_1$ and $\lambda_2$ as 0.1 and 1.0.

## B  The Proof for Theorem 1

**Theorem 1** (Data Quantity Impact). *Suppose in PAC Bayesian [40], for a target domain $\mathcal{T}$ and a source domain $\mathcal{S}$, any set of voters (candidate models) $\mathcal{H}$, any prior $\pi$ over $\mathcal{H}$ before any training, any $\xi \in (0, 1]$, any $c > 0$, with a probability at least $1 - \xi$ over the choices of $S \sim \mathcal{S}^{n_s}$ and $T \sim \mathcal{T}_{\mathcal{X}}^{n_t}$, for the posterior $f$ over $\mathcal{H}$ after the joint training on $S$ and $T$, we have*

$$\begin{aligned} \mathcal{R}_{\mathcal{T}}(f) \leq &\frac{c}{2(1-e^{-c})} \widehat{\mathcal{R}}_T(f) + \frac{c}{1-e^{-c}} \beta_\infty(\mathcal{T}\|\mathcal{S}) \widehat{\mathcal{R}}_S(f) + \Omega \\ &+ \frac{1}{1-e^{-c}} \left( \frac{1}{n_t} + \frac{\beta_\infty(\mathcal{T}\|\mathcal{S})}{n_s} \right) \left( 2\text{KL}(f\|\pi) + \ln\frac{2}{\xi} \right), \end{aligned} \tag{15}$$

*where $\widehat{\mathcal{R}}_T(f)$ and $\widehat{\mathcal{R}}_S(f)$ are the target and source empirical risks measured over target and source datasets $T$ and $S$, respectively. $\Omega$ is a constant and $\text{KL}(\cdot)$ is the Kullback–Leibler divergence. $\beta_\infty(\mathcal{T}\|\mathcal{S})$ is a measurement of discrepancy between $\mathcal{T}$ and $\mathcal{S}$ defined as*

$$\beta_\infty(\mathcal{T}\|\mathcal{S}) = \sup_{(\boldsymbol{x},y)\in\text{SUPP}(\mathcal{S})} \left( \frac{\mathcal{P}_{(\boldsymbol{x},y)\in\mathcal{T}}}{\mathcal{P}_{(\boldsymbol{x},y)\in\mathcal{S}}} \right) \geq 1, \tag{16}$$

*where $\text{SUPP}(\mathcal{S})$ denotes the support of $\mathcal{S}$. When $\mathcal{S}$ and $\mathcal{T}$ are identical, $\beta_\infty(\mathcal{T}\|\mathcal{S}) = 1$.*

*Proof.* Theorem 6 in Germain *et al.* 's work [55] demonstrates that *suppose in PAC Bayesian [40], for a target domain $\mathcal{T}$ and a source domain $\mathcal{S}$, any set of voters (candidate models) $\mathcal{H}$, any prior $\pi$ over $\mathcal{H}$ before any training, any $\xi \in (0,1]$, any $c > 0$, with a probability at least $1 - \xi$ over the choices of $S \sim \mathcal{S}^{n_s}$ and $T \sim \mathcal{T}_{\mathcal{X}}^{n_t}$, for the posterior $f$ over $\mathcal{H}$ after the joint training on $S$ and $T$:*

$$
\mathcal{R}_{\mathcal{T}}(f) \leq \frac{c}{2(1 - e^{-c})}\widehat{d}_T(f) + \frac{c}{1 - e^{-c}}\beta_{\infty}(\mathcal{T}\|\mathcal{S})\widehat{e}_S(f) + \Omega
$$
$$
+ \frac{1}{1 - e^{-c}}\left(\frac{1}{n_t} + \frac{\beta_{\infty}(\mathcal{T}\|\mathcal{S})}{n_s}\right)\left(2\mathrm{KL}(f\|\pi) + \ln\frac{2}{\xi}\right), \tag{17}
$$

*where $\mathcal{R}_{\mathcal{T}}(f)$ denotes the expected Gibbs risk of voter $f$ over the target domain. $\widehat{d}_T(f)$ and $\widehat{e}_S(f)$ are the empirical estimation of the target voters' disagreement and the source joint error, measured over target and source datasets $T$ and $S$, respectively. $\Omega$ is a constant and $\mathrm{KL}(\cdot)$ is the Kullback–Leibler divergence. $\beta_{\infty}(\mathcal{T}\|\mathcal{S})$ measures the discrepancy between $\mathcal{T}$ and $\mathcal{S}$, defined as:*

$$
\beta_{\infty}(\mathcal{T}\|\mathcal{S}) = \sup_{(\boldsymbol{x},y)\in\mathrm{SUPP}(\mathcal{S})}\left(\frac{\mathcal{P}_{(\boldsymbol{x},y)\in\mathcal{T}}}{\mathcal{P}_{(\boldsymbol{x},y)\in\mathcal{S}}}\right), \tag{18}
$$

*where $\mathrm{SUPP}(\mathcal{S})$ denotes the support of $\mathcal{S}$.*

In the following proof, in particular, the Gibbs risk $\mathcal{R}_{\mathcal{A}}(f)$, the voters' disagreement $d_{\mathcal{A}}(f)$, and the joint error $e_{\mathcal{A}}(f)$ of a certain domain $\mathcal{A}$ are defined as follows.

$$
\mathcal{R}_{\mathcal{A}}(f) = \mathop{\mathbb{E}}_{(\boldsymbol{x},y)\sim\mathcal{A}}\mathop{\mathbb{E}}_{h\sim f}\mathbb{I}[h(\boldsymbol{x}) \neq y], \tag{19}
$$

$$
d_{\mathcal{A}}(f) = \mathop{\mathbb{E}}_{\boldsymbol{x}\sim\mathcal{A}_{\mathcal{X}}}\mathop{\mathbb{E}}_{h\sim f}\mathop{\mathbb{E}}_{h'\sim f}\mathbb{I}\left[h(\boldsymbol{x}) \neq h'(\boldsymbol{x})\right], \tag{20}
$$

$$
e_{\mathcal{A}}(f) = \mathop{\mathbb{E}}_{(\boldsymbol{x},y)\sim\mathcal{A}}\mathop{\mathbb{E}}_{h\sim f}\mathop{\mathbb{E}}_{h'\sim f}\mathbb{I}\left[h(\boldsymbol{x}) \neq y\right]\mathbb{I}\left[h'(\boldsymbol{x}) \neq y\right], \tag{21}
$$

where $\mathbb{I}[\mathrm{True}] = 1$ if the inner condition is true, and otherwise $\mathbb{I}[\mathrm{False}] = 0$, and $\mathcal{A}_{\mathcal{X}}$ is the marginal distribution of domain $\mathcal{A}$. $h$ and $h'$ are votes sampled from the posterior distribution $f$ over $\mathcal{H}$. With these definitions, studies [56, 57] reveal a relationship among the Gibbs risk, the voters' disagreement, and the joint error as

$$
\mathcal{R}_{\mathcal{A}}(f) = \mathop{\mathbb{E}}_{(\boldsymbol{x},y)\sim\mathcal{A}}\mathop{\mathbb{E}}_{h\sim f}\mathop{\mathbb{E}}_{h'\sim f}\frac{\mathbb{I}\left[h(\boldsymbol{x}) \neq h'(\boldsymbol{x})\right] + 2\mathbb{I}\left[h(\boldsymbol{x}) \neq y \wedge h'(\boldsymbol{x}) \neq y\right]}{2} = \frac{1}{2}d_{\mathcal{A}}(f) + e_{\mathcal{A}}(f). \tag{22}
$$

In this case, we can extend this relationship to the empirical estimations (suppose a dataset $A$ is sampled from domain $\mathcal{A}$) as

$$
\widehat{\mathcal{R}}_A(f) = \frac{1}{|A|}\sum_{(\boldsymbol{x},y)\sim A}\mathop{\mathbb{E}}_{h\sim f}\mathop{\mathbb{E}}_{h'\sim f}\frac{\mathbb{I}\left[h(\boldsymbol{x}) \neq h'(\boldsymbol{x})\right] + 2\mathbb{I}\left[h(\boldsymbol{x}) \neq y \wedge h'(\boldsymbol{x}) \neq y\right]}{2} = \frac{1}{2}\widehat{d}_A(f) + \widehat{e}_A(f). \tag{23}
$$

Then we can use $\widehat{\mathcal{R}}_T(f)$ and $\widehat{\mathcal{R}}_S(f)$ to replace $\widehat{d}_T(f)$ and $\widehat{e}_S(f)$ in Eq. (17), respectively. In the end, we can follow Xu *et al.* [58] to regard these empirical risks as data quantity-irrelevant when analyzing the impact of data quantity.

Next, we focus on the proof of the numerical relationship $\beta_{\infty}(\mathcal{T}\|\mathcal{S}) \geq 1$. First of all, $\beta_{\infty}(\mathcal{T}\|\mathcal{S})$ comes from a more general definition that is parameterized by a real value $q > 0$, shown as

$$
\beta_q(\mathcal{T}\|\mathcal{S}) = \left[\mathop{\mathbb{E}}_{(\boldsymbol{x},y)\sim\mathcal{S}}\left(\frac{\mathcal{P}_{(\boldsymbol{x},y)\in\mathcal{T}}}{\mathcal{P}_{(\boldsymbol{x},y)\in\mathcal{S}}}\right)^q\right]^{\frac{1}{q}}. \tag{24}
$$

For any $q > 0$, $\beta_q(\mathcal{T}\|\mathcal{S})$ can be also written as a Ŕenyi Divergence-based form [55], *i.e.*,

$$
\beta_q(\mathcal{T}\|\mathcal{S}) = 2^{\frac{q-1}{q}D_q(\mathcal{T}\|\mathcal{S})}, \tag{25}
$$

where $D_q(\mathcal{T}\|\mathcal{S})$ is the Ŕenyi Divergence between $\mathcal{T}$ and $\mathcal{S}$ with the order $q$. For Ŕenyi Divergence with any order $q > 0$, there is a property of positivity [59], *i.e.*, $D_q(\mathcal{T}\|\mathcal{S}) \geq 0$. In this case, when $q \to \infty$, Eq. (25) becomes $\beta_q(\mathcal{T}\|\mathcal{S}) = 2^{D_q(\mathcal{T}\|\mathcal{S})} \geq 1$, and $\beta_q(\mathcal{T}\|\mathcal{S}) = 2^{D_q(\mathcal{T}\|\mathcal{S})} = 1$ when $\mathcal{T} = \mathcal{S}$, in other words, $D_q(\mathcal{T}\|\mathcal{S}) = 0$ when $\mathcal{T} = \mathcal{S}$ [59]. $\square$

# C    Technical Details for Generating Protected Dataset

## C.1    The Optimization Solution for Generating Protected Dataset

Recall that Eq. (7) is formulated as follows.

$$\min_{\boldsymbol{\delta} \subset \mathcal{B}} \left[ \mathbb{E}_{(\hat{\boldsymbol{x}},y)\sim\mathcal{T}}[\mathcal{L}\left(f(\hat{\boldsymbol{x}};\boldsymbol{w}(\boldsymbol{\delta})),y\right)] - \lambda_3 \min\left\{ \mathbb{E}_{(\overline{\boldsymbol{x}},y)\sim\overline{\mathcal{T}}}[\mathcal{L}\left(f(\overline{\boldsymbol{x}};\boldsymbol{w}(\boldsymbol{\delta})),y\right)], \lambda_4 \right\} \right], \tag{26}$$

$$s.t.\ \boldsymbol{w}(\boldsymbol{\delta}) = \arg\min_{\boldsymbol{w}} \left[ \frac{1}{|\mathcal{D}_s|} \sum_{(\boldsymbol{x}_i,y_i)\in\mathcal{D}_s} \mathcal{L}\left(f(\boldsymbol{x}_i + \boldsymbol{\delta}_i; \boldsymbol{w}), y_i\right) + \frac{1}{|\mathcal{D}_b|} \sum_{(\boldsymbol{x}_j,y_j)\in\mathcal{D}_b} \mathcal{L}\left(f(\boldsymbol{x}_j;\boldsymbol{w}), y_j\right) \right],$$

where $\mathbb{E}_{(\overline{\boldsymbol{x}},y)\sim\overline{\mathcal{T}}}[\mathcal{L}\left(f(\overline{\boldsymbol{x}};\boldsymbol{w}(\boldsymbol{\delta})),y\right)]$ represents the expected risk for the watermarked model on other unseen domains (*i.e.*, $\overline{\mathcal{T}}$) and $\mathcal{B} = \{\boldsymbol{\delta} : ||\boldsymbol{\delta}||_\infty \leq \epsilon\}$ where $\epsilon$ is a visibility-related hyper-parameter.

The aforementioned problem is a standard bi-level problem, we following previous work [60, 61] to leverage *gradient matching* to solving it. Specifically, we first make the following definition:

$$\mathcal{L}_t = \mathbb{E}_{(\hat{\boldsymbol{x}},y)\sim\mathcal{T}}[\mathcal{L}\left(f(\hat{\boldsymbol{x}};\boldsymbol{w}),y\right)] - \lambda_3 \min\left\{ \mathbb{E}_{(\overline{\boldsymbol{x}},y)\sim\overline{\mathcal{T}}}[\mathcal{L}\left(f(\overline{\boldsymbol{x}};\boldsymbol{w}),y\right)], \lambda_4 \right\}, \tag{27}$$

$$\mathcal{L}_i = \frac{1}{|\mathcal{D}_s|} \sum_{(\boldsymbol{x}_i,y_i)\in\mathcal{D}_s} \mathcal{L}\left(f(\boldsymbol{x}_i + \boldsymbol{\delta}_i; \boldsymbol{w}), y_i\right). \tag{28}$$

According to the gradient-matching technique [60, 61], we have the Upper-level Sub-problem as:

$$\max_{\boldsymbol{\delta} \subset \mathcal{B}} \frac{\nabla_{\boldsymbol{w}} \mathcal{L}_t \cdot \nabla_{\boldsymbol{w}} \mathcal{L}_i}{|| \nabla_{\boldsymbol{w}} \mathcal{L}_t || \cdot || \nabla_{\boldsymbol{w}} \mathcal{L}_i ||}, \tag{29}$$

where we aim to maximize the gradient matching degree between $\nabla_{\boldsymbol{w}}\mathcal{L}_t$ and $\nabla_{\boldsymbol{w}}\mathcal{L}_i$ using $\texttt{cosine}(\cdot)$ similarity as the metric through optimizing $\delta$. We solve the above upper-level sub-problem via projected gradient ascend. We here use calculate $\mathbb{E}_{(\hat{\boldsymbol{x}},y)\sim\mathcal{T}}[\mathcal{L}\left(f(\hat{\boldsymbol{x}};\boldsymbol{w}),y\right)]$ following:

$$\mathbb{E}_{(\hat{\boldsymbol{x}},y)\sim\mathcal{T}}[\mathcal{L}\left(f(\hat{\boldsymbol{x}};\boldsymbol{w}),y\right)] = \frac{1}{N} \sum_{(\boldsymbol{x},y)\in\mathcal{D}} \mathcal{L}(f(G_d(\boldsymbol{x});\boldsymbol{w}),y). \tag{30}$$

Regarding the lower-level sub-problem, we have:

$$\min_{\boldsymbol{w}} \left[ \frac{1}{|\mathcal{D}_s|} \sum_{(\boldsymbol{x}_i,y_i)\in\mathcal{D}_s} \mathcal{L}\left(f(\boldsymbol{x}_i + \boldsymbol{\delta}_i; \boldsymbol{w}), y_i\right) + \frac{1}{|\mathcal{D}_b|} \sum_{(\boldsymbol{x}_j,y_j)\in\mathcal{D}_b} \mathcal{L}\left(f(\boldsymbol{x}_j;\boldsymbol{w}), y_j\right) \right]. \tag{31}$$

After obtaining the poisoned dataset (*i.e.*, $\mathcal{D}_s \cup \mathcal{D}_b$), we can optimize the model (*i.e.*, ResNet-18) parameters $\boldsymbol{w}$ via solving the above Lower-level Upper-sub problem. The above Lower-level Upper-sub problem is solved via stochastic gradient descent.

We optimize the Upper-level and Lower-level Sub-problems alternatively for each optimization iteration. Specifically, we first train the model under benign dataset $\mathcal{D}$. Then for each iteration, we first optimize the Upper-level Sub-problem based on the trained model and obtain the perturbation $\delta$. After that, we optimize the Lower-level Sub-problem based on the obtained poisoned dataset. During each iteration for optimizing the above bi-level optimization problems, we optimize the Upper-level Sub-problem with 50 iterations, and optimize the Lower-level Sub-problem with 100 iterations. We optimize the entire bi-level optimization with five epochs. The other details for optimization hyper-parameters as well as configuration are consistent with [61, 29].

In particular, to ensure the effectiveness of solving the aforementioned bi-level optimization problem, we have two additional strategies, as follows:

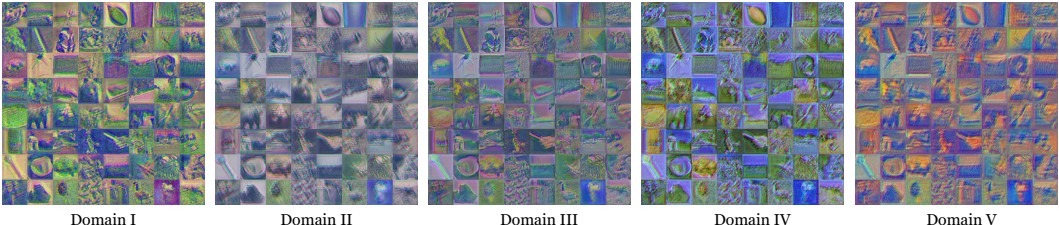

| Domain I | Domain II | Domain III | Domain IV | Domain V |

Figure 10: The example of samples generated from various domain.

- **Strategy 1:** Instead of randomly selecting samples from benign dataset $\mathcal{D}$, we here choose to select training samples with the largest gradient norms, following the previous work [60].
- **Strategy 2:** Instead of selecting samples from all classes, we follow the previous work [4] to select those from a specific class and the selected class is set as the target label. This strategy can enhance the effectiveness for solving the above bi-level optimization problem while preserving the verification performance for our approach.

## C.2 The Process of Generating Samples from Other Domains

In this part, we describe how to generate samples from other domains (*i.e.*, $(\overline{\boldsymbol{x}}, y) \sim \overline{\mathcal{T}}$).

After obtaining the transformation module $G_d(\cdot)$, we can generate hard-generalized domain samples from a specific domain. We here propose to generate samples from other domains by setting different configurations of $\{w_i\}_1^4$. For example, we can generate samples from the other domain by sampling $\{w_i\}_1^4$ with another values following $w_i \sim N(0, 1)$.

We here show some demonstration of samples from other domains in Fig. 10.

We here generated samples from other domains, and estimate $\mathbb{E}_{(\overline{\boldsymbol{x}}, y) \sim \overline{\mathcal{T}}}[\mathcal{L}\left(f(\overline{\boldsymbol{x}}; \boldsymbol{w}), y\right)]$ following:

$$\mathbb{E}_{(\overline{\boldsymbol{x}}, y) \sim \overline{\mathcal{T}}}[\mathcal{L}\left(f(\overline{\boldsymbol{x}}; \boldsymbol{w}(\boldsymbol{\delta})), y\right)] = \frac{1}{N}\frac{1}{J}\sum_{j}\sum_{(\overline{\boldsymbol{x}}, \boldsymbol{y}) \in \overline{\mathcal{T}}_j} \mathcal{L}(f(\overline{\boldsymbol{x}}; \boldsymbol{w}), y), \tag{32}$$

where $\overline{\mathcal{T}}_j$ represents the $i$-th unseen domain generated by the above approach.

## C.3 The Selection of Hyper-parameters

After generating other unseen domains $\mathcal{T}$, we here describe the selection of hyper-parameters (*i.e.*, $J$ and $\lambda_3$) for generating protected dataset.

We here propose a heuristic approach for selecting $J$ and $\lambda_3$. Specifically, we first keep $\lambda_3$ fixed (*i.e.*,1) and adjust $J$. We conduct empirical study on CIFAR-10 tasks, the results are shown in Fig. 11.

We use ResNet-18 as the evaluated model. We generate several unseen domains using the above approach. We randomly select $J$ of these domains for optimizing the Eq. (7), and select 3 unseen domains as the validation data. Notably, the validation domains are ensured visually different from the domains used for optimization.

From Fig. 11, we find that using $\geq$ three unseen domains is sufficient to constrain the generalization performance for validation unseen domains. Therefore, we set $J$ as 3 for our approach.

After that, we keep $J$ fixed, and adjust $\lambda_3$ gradually, the results are shown in Fig. 12. We find that when $\lambda_3$ becomes smaller, the constraint for performance on other unseen domains reduces. Accordingly, we set $\lambda_3$ as 0.3 for our approach since it can achieve a close generalization capacity compared to the benign DNN model (*i.e.*, 24.3%).

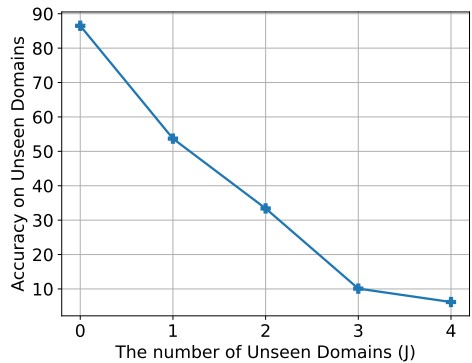

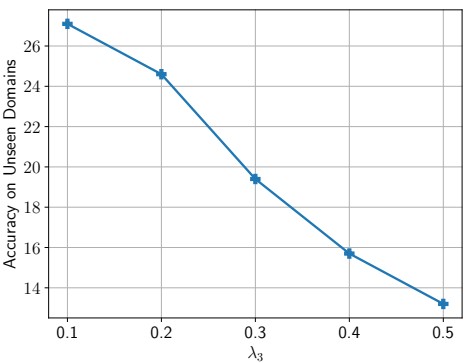

Figure 11: Effects of the number of unseen Domains $J$.

Figure 12: Effects of the $\lambda_3$.

## D  The Proof for Theorem 2

**Theorem 2.** *Let $f(\boldsymbol{x})$ is the posterior probability of $\boldsymbol{x}$ predicted by the suspicious model, variable $\boldsymbol{X}$ denotes the benign sample with label $Y$, and variable $\boldsymbol{X}'$ is the domain-watermarked version of $\boldsymbol{X}$. Assume that $P_b \triangleq f(\boldsymbol{X})_Y > \eta$. We claim that dataset owners can reject the null hypothesis $H_0$ at the significance level $\alpha$, if the verification success rate (VSR) $V$ of $f$ satisfies that*

$$\sqrt{m-1} \cdot (V - \eta + \tau) - t_\alpha \cdot \sqrt{V - V^2} > 0, \tag{33}$$

*where $t_\alpha$ is $\alpha$-quantile of t-distribution with $(m-1)$ degrees of freedom and $m$ is sample size.*

*Proof.* Since $\boldsymbol{P}_b > \eta$, the original hypothesis $H_1$ can be converted to

$$H_1' : P_d > \eta - \tau. \tag{34}$$

Let $E$ indicates the event of whether the suspect model $f$ predicts a watermark sample as its ground-truth label $y$. As such, $E \sim B(1, p)$, where $p = \Pr(C(\boldsymbol{X}') = Y)$ indicates the verification success probability and $B$ is the Binomial distribution [41].

Let $\hat{\boldsymbol{x}}_1, \cdots, \hat{\boldsymbol{x}}_m$ denotes $m$ domain-watermarked samples used for dataset verification and $E_1, \cdots, E_m$ denote their prediction events, we know that the verification success rate $V$ satisfies

$$V = \frac{1}{m} \sum_{i=1}^{m} E_i, \tag{35}$$

$$V \sim \frac{1}{m} B(m, p). \tag{36}$$

According to the central limit theorem [41], the verification success rate $V$ follows Gaussian distribution $\mathcal{N}(p, \frac{p(1-p)}{m})$ when $m$ is sufficiently large. Similarly, $(P_d - \eta + \tau)$ also satisfies Gaussian distribution. Accordingly, we can construct the t-statistic as follows:

$$T \triangleq \frac{\sqrt{m}(W - \eta + \tau)}{s} \sim t(m-1), \tag{37}$$

where $s$ is the standard deviation of $(V - \eta + \tau)$ and $V$, *i.e.*,

$$s^2 = \frac{1}{m-1} \sum_{i=1}^{m} (E_i - V)^2 = \frac{1}{m-1}(m \cdot V - m \cdot V^2). \tag{38}$$

To reject the hypothesis $H_0$ at the significance level $\alpha$, we need to ensure that

$$\frac{\sqrt{m}(V - \eta + \tau)}{s} > t_\alpha, \tag{39}$$

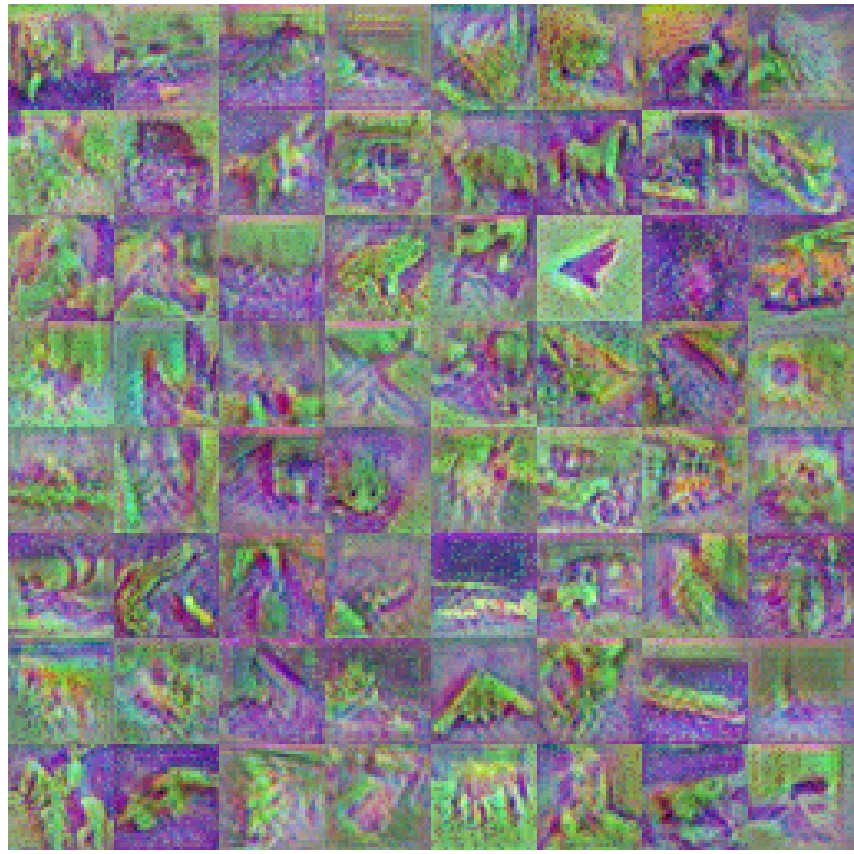

Figure 13: The example of domain watermark for CIFAR-10.

where $t_\alpha$ is the $\alpha$-quantile of t-distribution with $(m-1)$ degrees of freedom.

According to equation (38)-(39), we have

$$\sqrt{m-1} \cdot (V - \eta + \tau) - t_\alpha \cdot \sqrt{V - V^2} > 0. \tag{40}$$

$\square$

# E    The Detailed Settings for Experimental Datasets and Configurations

## E.1    Datasets

We evaluate our approach on three benchmark datasets (*i.e.*, CIFAR-10 [1], Tiny-ImgaeNet [42], and STL-10 [45]). We here describe each benchmark dataset in detail.

**CIFAR-10.**    CIFAR-10 dataset contains 10 labels, 50,000 training samples, and 10,000 validation samples. The training and validation samples are distributed evenly across each label. Each sample is resized as $32 \times 32$ by default.

**Tiny-ImageNet.**    Tiny-ImageNet dataset contains 200 labels, 100,000 training samples, and 10,000 validation samples. The training and validation samples are distributed evenly across each label. Each sample is resized as $64 \times 64$ by default.

**STL-10.**    STL-10 dataset contains 10 labels and 13,000 labeled samples and 100,000 unlabeled samples. We divide the labeled samples into the training and validation dataset with a ratio of $8 : 2$. The training and validation samples are distributed evenly across each label. Each sample is resized as $96 \times 96$ by default.

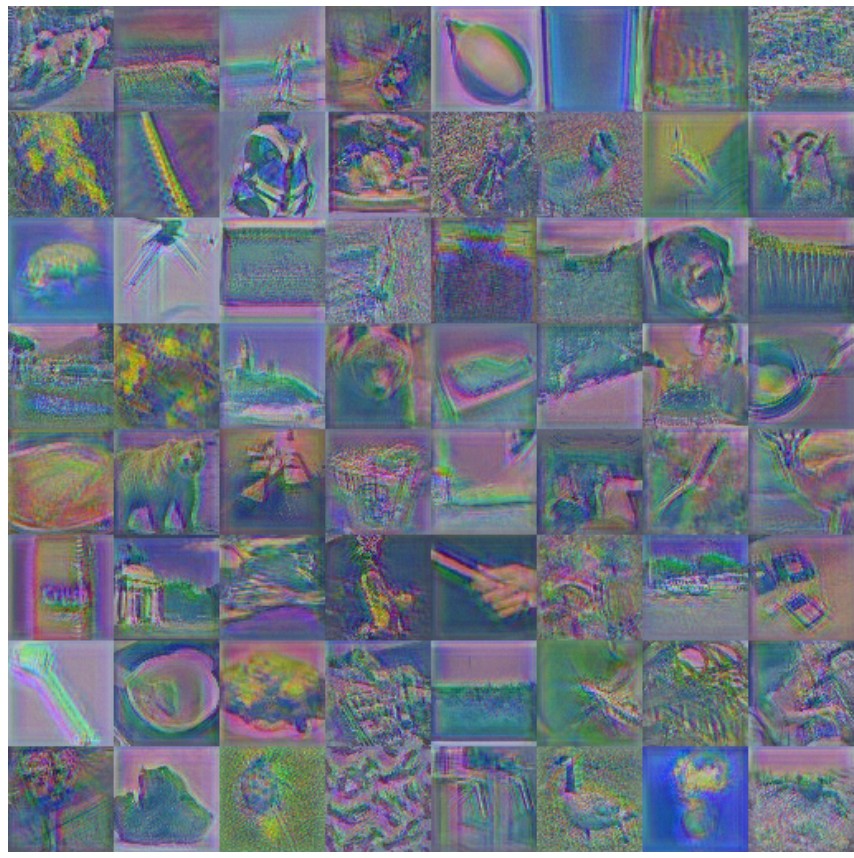

Figure 14: The example of domain watermark for Tiny-ImageNet.

Table 4: Summary of accuracy (%) on samples from different domains for normal models and ours.

| Task | Source domain | | Target domain | | Other domain | |
|---|---|---|---|---|---|---|
| | Normal | Ours | Normal | Ours | Normal | Ours |
| CIFAR-10 | 91.89 | 90.86 | 13.10 | 90.45 | 15.10 | 10.30 |
| STL-10 | 85.61 | 84.58 | 9.50 | 82.00 | 16.00 | 11.60 |
| Tiny-ImageNet | 60.13 | 59.10 | 6.00 | 58.08 | 12.60 | 15.40 |

## E.2 The Demonstration of Domain Watermark for Each Dataset

We here show the domain watermark used for evaluating the effectiveness of our approach in the experiments. The demonstrations are shown in Figure 13, Figure 14, and Figure 15 for CIFAR-10, Tiny-ImageNet, and STL-10 datasets, respectively.

## E.3 Training Configurations

In the experiments, we train each model with 150 epochs with an initialized learning rate of 0.1. Following previous work [29, 62], we schedule learning rate drops at epochs 14, 24, and 35 by a factor of 0.1. For all models, we employ SGD with Nesterov momentum, and we set the momentum coefficient to 0.9. We use batches of 128 images and weight decay with a coefficient of $4 \times 10^{-4}$. For each run, we report the verification success rate (VSR) averaged over the last 10 epochs when the models' accuracy converges. We report the results for each approach averaged over 5 runs.

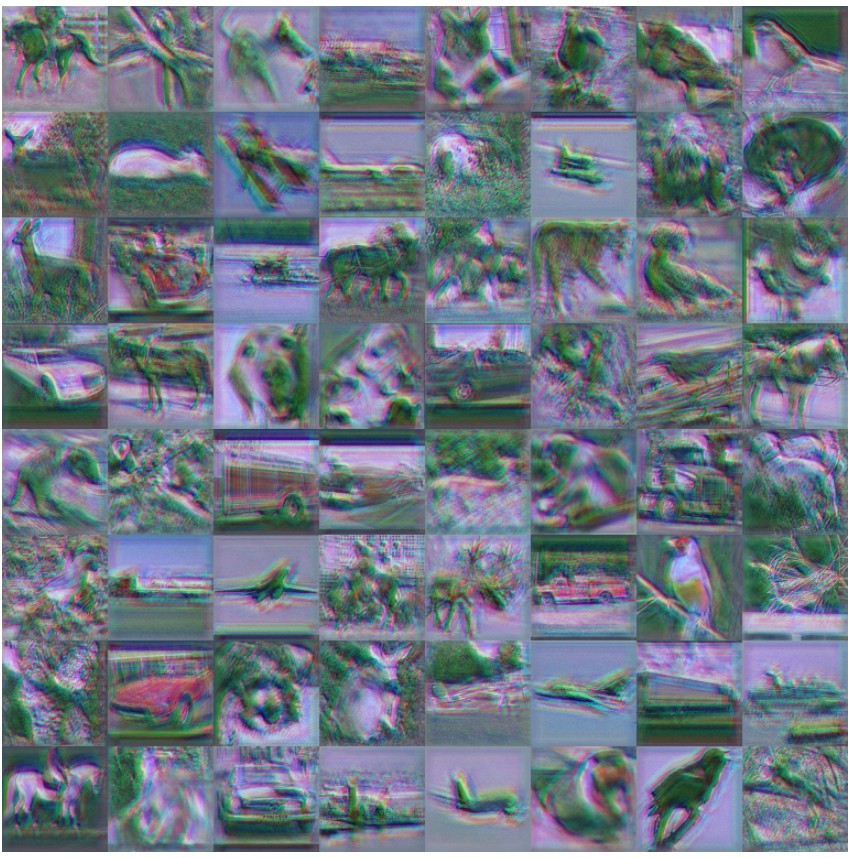

Figure 15: The example of domain watermark for STL-10.

Table 5: The watermark performance on STL-10 dataset. In particular, we mark harmful watermark results (*i.e.*, $H > 0.5$ and $\hat{H} > 0$) in red.

| Label Type↓ | Method↓, Metric→ | BA (%) | VSR (%) | $H$ | $\hat{H}$ |
|---|---|---|---|---|---|
| | | | STL-10 | | |
| Poisoned-Label | BadNets | 85.61 | 100 | 1.00 | 0.86 |
| | Blended | 85.21 | 99.32 | 1.00 | 0.84 |
| | WaNet | 83.17 | 96.10 | 0.96 | 0.79 |
| | UBW-P | 84.22 | 80.27 | 0.80 | 0.64 |
| Clean-Label | Label-Consistent | 84.07 | 93.48 | 0.93 | 0.77 |
| | Sleeper Agent | 83.72 | 89.77 | 0.90 | 0.73 |
| | UBW-C | 79.32 | 82.00 | 0.82 | 0.61 |
| | DW (Ours) | 84.58 | 82.00 | 0.18 | -0.73 |

## E.4 The Details for Implementing each Approach

We implement each backdoor technique using `Backdoorbox` library[3] following the default training configurations. Specifically, for patch-based triggers, we use $3 \times 3$, $6 \times 6$, and $9 \times 9$ for CIFAR-10, Tiny-ImageNet, and STL-10. Following previouw work [4], for each approach, we randomly select a label as the target label for ownership verification purposes. For the other input-specific trigger (*i.e.*, WaNet [27]), we follow its default configuration to generate its specific trigger pattern.

---

[3] https://github.com/THUYimingLi/BackdoorBox

Table 6: The effectiveness of dataset ownership verification via our domain watermark.

| | STL-10 | | |
|---|---|---|---|
| | Independent-D | Independent-M | Malicious |
| $\Delta P$ | 0.68 | 0.78 | 0.04 |
| p-value | 0.95 | 0.98 | $10^{-46}$ |

Table 7: Summary of accuracy (%) on samples from different domains for normal models and ours.

| Domain Watermark | Source domain | | Target domain | | Other domain | |
|---|---|---|---|---|---|---|
| | Normal | Ours | Normal | Ours | Normal | Ours |
| Domain Watermark I | 92.46 | 92.10 | 18.50 | 91.40 | 16.30 | 17.60 |
| Domain Watermark II | 92.46 | 91.95 | 18.20 | 90.24 | 14.70 | 15.80 |
| Domain Watermark III | 92.46 | 91.85 | 19.60 | 90.64 | 18.40 | 14.90 |

## F The Additional Results for the Performance of Domain Watermark

We first show the summary for the performance of our approach and benign samples on samples from different domains. The results are shown in Table 4. We also show additional results for STL-10 dataset with ResNet-34 as shown in Table 5.

## G The Detailed Settings for Dataset Ownership Verification

We evaluate our domain-watermark-based dataset ownership verification under three scenarios, including **1)** independent domain (dubbed 'Independent-D'), **2)** independent model (dubbed 'Independent-M'), and **3)** unauthorized dataset training (dubbed 'Malicious'). In the first case, we used domain-watermarked samples to query the suspicious model trained with modified samples from another domain; In the second case, we test the benign model with our domain-watermarked samples; In the last case, we test the domain-watermarked model with corresponding domain-watermarked samples. Notice that only the last case should be regarded as having unauthorized dataset adoption. All other settings are the same as those used in [4].

Consistent with previouw work [4], we adopt the trigger used in the training process of the watermarked suspicious model in the last scenario. Moreover, we sample $m = 100$ samples on CIFAR10, STL-10, and Tiny-ImageNet and set $\tau = 0.25$ for the hypothesis-test in each case for our approach. Since Tiny-ImageNet has only 50 samples for each class in the validation dataset, we combine additional 50 training samples with the validation samples for ownership verification. The additional 50 training samples are not used in generating the protected dataset.

## H The Additional Results for Dataset Ownership Verification

We here investigate the effectiveness of ownership verification via our domain watermark. The results are shown in Table 6. The settings are consistent with Section 5.

## I Additional Results of Discussions

### I.1 The Effects of $\lambda_3$

We have investigated the effects of $\lambda_3$, as shown in Fig. 12. We find that the generalization performance decreases on other unseen validation domains with the increase of $\lambda_3$. When $\lambda_3$ increases up to 0.3, the generalization performance on other unseen validation domains decreases close to the generalization performance for benign models.

### I.2 Performance under Different Domain Watermarks

We here investigate the effective of protected dataset generation for different domain watermarks. We here craft domain watermarks following the Appendix A but initialized with different parameters

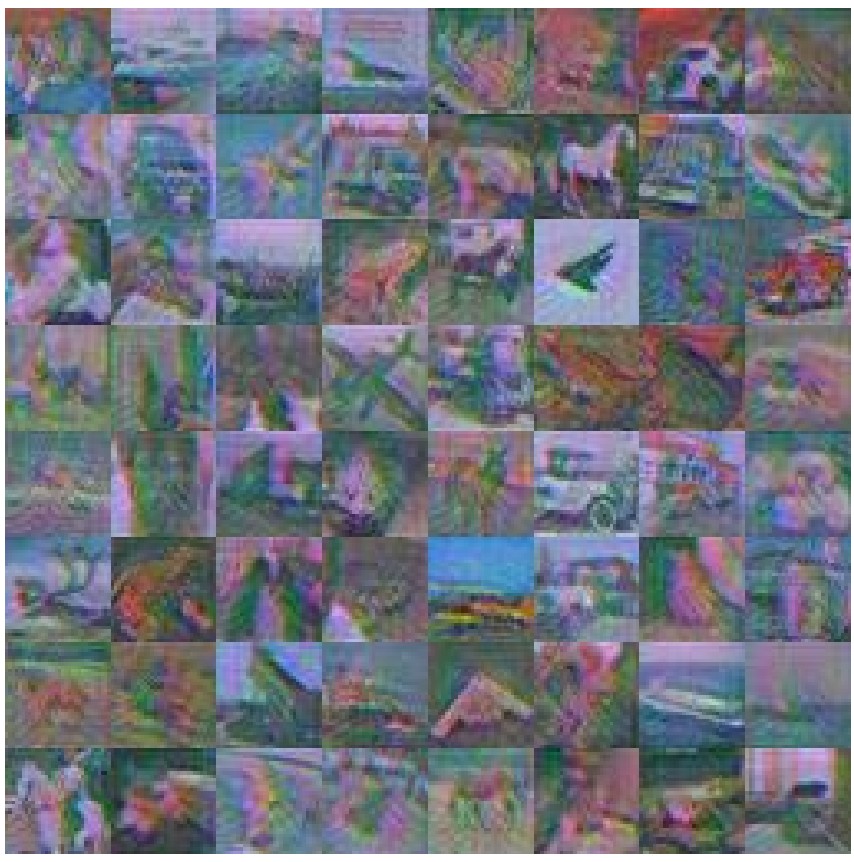

Figure 16: The Demonstration of Domain Watermark I.

Table 8: The performance of our *domain watermark* with different model structures trained on the watermarked dataset generated with ResNet-18.

| Metric↓, Model→ | ResNet-18 | ResNet-34 | VGG-16-BN | VGG-19-BN |
|---|---|---|---|---|
| BA (%) | 91.39 | 92.54 | 90.86 | 92.57 |
| VSR (%) | 91.90 | 90.80 | 90.48 | 89.00 |

for crafting different domain watermarks. The demonstrations for different domain watermarks for CIFAR-10 are shown in Figs. 16 to 18.

We here use CIFAR-10 with ResNet-34 to investigate the performance of our approach for different domain watermarks. The results are summarized in Tab. 7. We can see our approach can still achieve effectiveness for different domain watermarks.

### I.3    The Transferability of Domain Watermark

Recall that in the optimization process of our approach, we leverage a surrogate model (*i.e.*, ResNet-18) for crafting modified samples. In the experiment section, we test the effectiveness of our approach under models (*i.e.*, VGG-16-BN and ResNet-34) having different architectures and parameters from the surrogate model. In practice, dataset users may adopt different model structures since dataset owners have no information about the model training. In this section, we conduct additional experiments on evaluating the effectiveness of our approach under different structures compared to the one used for generating modified samples (*i.e.*, transferability).

**Settings.**    We evaluate the transferability of our method on CIFAR-10. We adopt ResNet-18, ResNet-34, VGG-16-BN, and VGG-19-BN to peform domain watermark, based on which to train

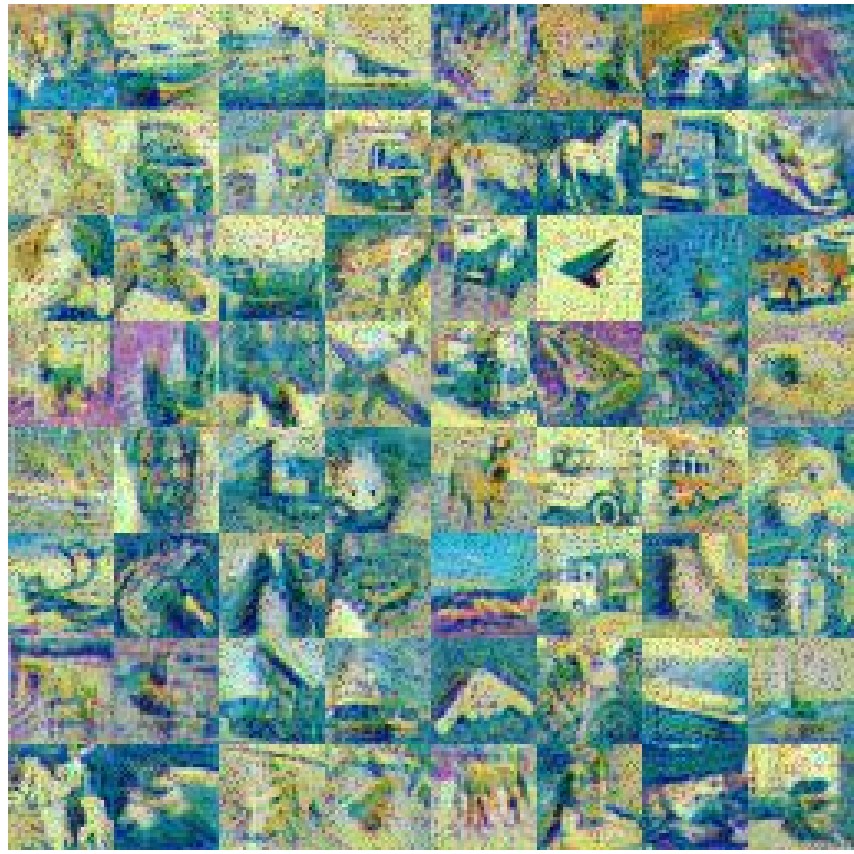

Figure 17: The Demonstration of Domain Watermark II.

Table 9: The performance of our *domain watermark* with different model structures trained on the watermarked dataset generated with ResNet-34.

| Metric↓, Model→ | ResNet-18 | ResNet-34 | VGG-16-BN | VGG-19-BN |
|---|---|---|---|---|
| BA (%) | 91.22 | 92.56 | 90.43 | 91.79 |
| VSR (%) | 90.10 | 92.44 | 89.60 | 90.36 |

Table 10: The performance of our *domain watermark* with different model structures trained on the watermarked dataset generated with VGG-16-BN.

| Metric↓, Model→ | ResNet-18 | ResNet-34 | VGG-16-BN | VGG-19-BN |
|---|---|---|---|---|
| BA (%) | 91.57 | 92.10 | 90.53 | 92.10 |
| VSR (%) | 90.70 | 91.60 | 90.44 | 89.84 |

Table 11: The performance of our *domain watermark* with different model structures trained on the watermarked dataset generated with VGG-19-BN.

| Metric↓, Model→ | ResNet-18 | ResNet-34 | VGG-16-BN | VGG-19-BN |
|---|---|---|---|---|
| BA (%) | 91.48 | 91.98 | 90.77 | 92.73 |
| VSR (%) | 91.30 | 89.60 | 90.36 | 91.94 |

different models (*i.e.*, ResNet-18, ResNet-34, VGG-16-BN, and VGG-19-BN). Except for the model structure, all other settings are the same as those used in Section 5.

**Results.** As shown in Table 8-11, our method has high transferability across model structures. Accordingly, our methods are practical in protecting open-sourced datasets.

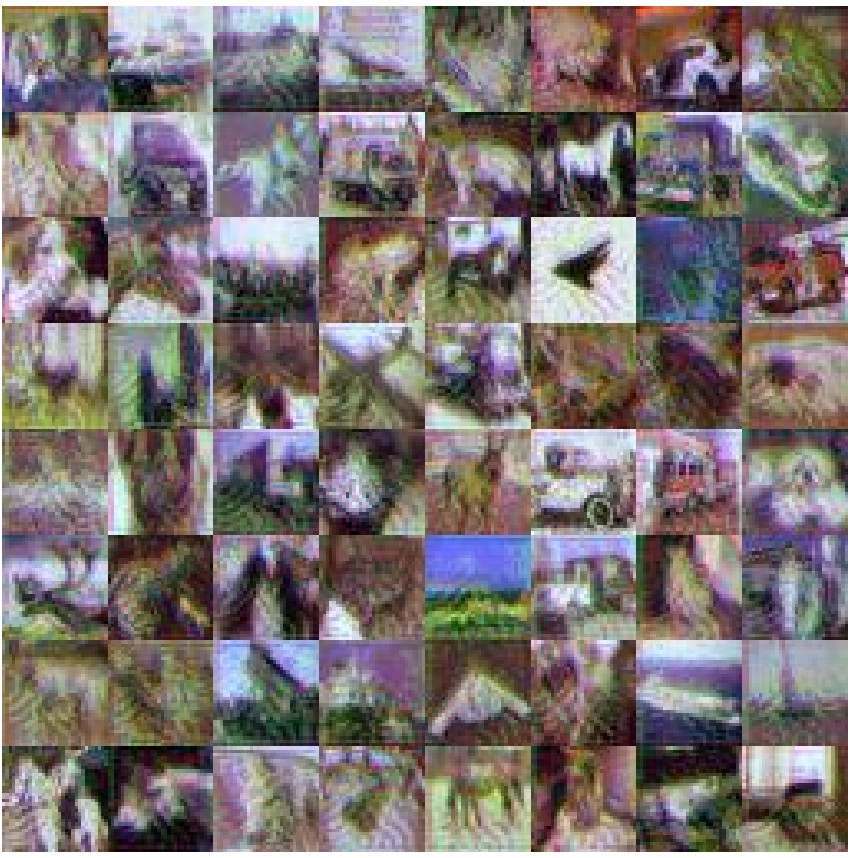

Figure 18: The Demonstration of Domain Watermark III.

## J  Additional Results for the Resistance to Potential Adaptive Methods

**Robustness against ShrinkPad.**  We hereby discuss the robustness of our method against ShrinkPad [63], which is a well-known watermarked sample detection approach based on a set of input transforamtions. We follow `BackdoorBox` to implement ShrinkPad for filtering watermarked samples. We use CIFAR-10 with ResNet-34 to implement *domain watermark* and craft 1,000 watermarked samples based on the validation dataset for investigation. We first filter 900 watermarked samples that can be correctly classified. We find ShrinkPad can only filter 87 effective watermarked samples among 900 samples ($\leq 10\%$), *i.e.*, our domain watermark is robust against ShrinkPad.

**Robustness against Scale-UP.**  We also evaluate our method under the most advanced input-level watermark detection (*i.e.*, SCALE-UP [64]). We follow their released code[4] to implement SCALE-UP and use the AUROC score as the metric to report the results. We test our approach on SCALE-UP with 1,000 watermarked and 1,000 benign samples. We here use CIFAR-10 with ResNet-34.

We find that SCALE-UP yields around $0.58$ AUROC score on our proposed *domain watermark*. Such results imply that SCALE-UP can not perform against our domain watermark, with the filtering performance close to random guesses. We think it may be caused by that, different from the previous backdoor-inspired watermark causing misclassification, *domain watermark* leads the watermarked model correctly classifying the watermarked samples. Therefore, the watermarked samples would have a similar scaled prediction consistency as benign samples, since they all belong to the ground-truth label and can be clustered closely as shown in Section 5.3.2.

**Robustness against Neural Cleanse.**  Following previous work [60], we also evaluate our approach against reverse-engineering based approaches [65, 66, 67].We here evaluate our approach

---

[4]https://github.com/JunfengGo/SCALE-UP

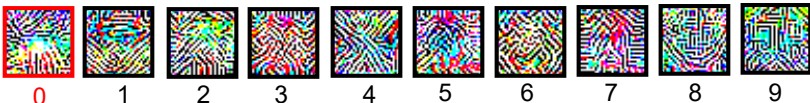

Figure 19: The reversed trigger maps for each label produced by Neural Cleanse.

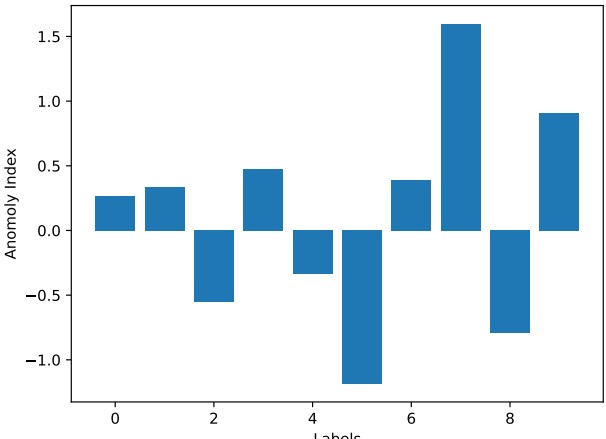

Figure 20: The anomaly index for $\ell_1$-norm computed on the reversed trigger maps for each label produced by Neural Cleanse.

against Neural Cleanse [67], which is the most widely-adopted approach. We select label 0 as the target label and use CIFAR-10 with ResNet-34. The results are shown in Figure 19-20. We can see the reversed trigger pattern produced by Neural Cleanse for the target label is extremely dense. We further follow [67] to calculate the anomaly index for each label using MAD outlier detection. We find that the target label's anomaly index is smaller than 2, thus it would not be detected.

We notice that there are still many other backdoor defenses (*e.g.*, [68, 69, 70]). We will discuss the resistance of our domain watermark method to them in our future works.

## K    Can We Identify Domain-watermarked Samples by Training on a Large Dataset or with Complicated Data Augmentation?

In general, our method is developed on the foundation that only models trained on our protected dataset can successfully identify domain-watermarked samples. However, people may worry about whether the suspicious model can also generalize well on domain-watermarked samples because its good training strategies or structures instead of because being trained on the protected dataset. In this section, we discuss this potential problem.

Firstly, we cannot ensure this problem will never happen since training with more samples and complicated data augmentations may indeed increase the general generalizability of suspicious models. However, the probability of it happening is very small in practice, as follows.

- Hardly-generalized domain is an exclusive and special domain where only defenders know what it is. Accordingly, the adversaries (*i.e.*, malicious dataset users) cannot know it and use it to break our verification.

- It is unlikely that a benign user can develop data augmentation that can generate (sufficiently) similar hard samples without knowing our specific settings, since the space of hardly-generalized domain is huge and our watermark is sample-specific and not unique.

- As shown in Figures 2-3 in the main manuscript as well as Figures 5-10 in the Appendix, domain-watermarked samples and not natural and very special. In other words, it is impossible that dataset users can collect them without using our protected dataset. Accordingly,

Table 12: The performance of training with strong data augmentations on the benign dataset.

| Dataset↓, Metric→ | BA (%) | Accuracy on DW samples (%) | $\Delta P$ | p-value |
|---|---|---|---|---|
| CIFAR-10 | 91.50 | 30.15 | 0.76 | 1.00 |
| Tiny-ImageNet | 75.84 | 6.00 | 0.63 | 1.00 |

    it is not reasonable to claim that a model has good performance on domain-watermarked samples because their dataset is extremely large covering numerous hard samples.

- Lemma 1 indicates that we can find a hardly-generalized domain for the dataset, no matter what the model architecture is. Accordingly, it is not reasonable to claim that a model has good performance on domain-watermarked samples due to its well-designed architecture.

- Currently, no model can have similar generalizability to humans, although it was trained on a huge dataset with complicated data augmentation strategies and a well-designed structure.

- Domain generalization is still an important yet unsolved problem in computer vision. Accordingly, there is no existing method can make DNNs generalize to all unseen domains (with performance on par to that of the source domain) for breaking our method.

- Our hypothesis-testing-based verification process ensures that suspicious models will be treated as trained on our protected dataset only when they have sufficiently high (not just certain) generalizability over a large number (not just a small number) of hard samples. Accordingly, our method can reduce the side effects of randomness to a large extent.

We admit that using data augmentation or other domain adaption techniques may increase the generalization ability of trained DNNs on domain-watermarked samples, although adversaries have no information of hardly-generalized domains and those augmented samples used for training are significantly different from those of domain-watermarked ones. However, this improvement is limited since these adapted domains are significantly different from the hardly-generalized domain (the domain space is high-dimensional and enormous). Accordingly, this mild improvement cannot break our method. To further verify it, we evaluate our method on models with classical automatic data augmentations (*e.g.*, color shifting) and domain adaption in Appendix K.1-K.2.

In particular, it does not diminish the practicality of our method, even if this rare event is likely to happen. Using verification-based defenses (in a legal system) requires an official institute for arbitration. Specifically, all commercial models should be registered here, based on the unique identification (*e.g.*, MD5 code) of their model files and training datasets, before being used online. When this official institute is established, its staff should take responsibility for the verification process. For example, they can require the company to provide the dataset with the same registered identification and then check whether it contains our protected samples (via algorithms). Of course, if the suspect model is proven to be benign, the user will need to pay a settlement to its company to prevent casual malicious ownership verification. In this case, even if our method misclassifies in rare cases, it does not compromise the interest of the suspect model. Our method is practical in this realistic situation, as long as it has a sufficiently high probability of correct verification.

### K.1 The Resistance to Data Augmentation

**Settings.** We adopt strong data augmentations to train models on benign datasets with the same settings used in our paper. Specifically, we adopt random flip, random rotation, random affine transformations, random color shifting as the augmentations on CIFAR-10 and Tiny-ImageNet datasets.

**Results.** As shown in Table 12, our method will not misjudge (p-value $\gg$ 0.05), although data augmentation can (slightly) increase the accuracy on our domain-watermarked samples. These results verify that our method is resistant to data augmentations.

### K.2 The Resistance to Domain Adaption

**Settings.** In this part, we adopt L2D [50] as the representative method to investigate the robustness of our domain watermark against domain adaption techniques. Specifically, we train both the watermarked and benign DNNs with L2D on CIFAR-10 and Tiny-ImageNet datasets.

Table 13: The performance of our domain watermark under training with L2D.

| Dataset↓, Metric→ | BA (%) | VSR (%) | Accuracy on Other DW samples (%) | $H$ | $\hat{H}$ |
|---|---|---|---|---|---|
| CIFAR-10 | 91.10 | 86.10 | 19.10 | -0.14 | -0.73 |
| Tiny-ImageNet | 48.63 | 42.00 | 16.00 | 0.58 | -0.26 |

Table 14: The verification performance via our domain watermark under training with L2D.

| | CIFAR-10 | | | Tiny-ImageNet | | |
|---|---|---|---|---|---|---|
| | Independent-D | Independent-M | Malicious | Independent-D | Independent-M | Malicious |
| $\Delta P$ | 0.63 | 0.60 | 0.05 | 0.53 | 0.53 | 0.07 |
| p-value | 0.90 | 0.71 | $10^{-33}$ | 0.96 | 0.95 | $10^{-12}$ |

**Results.** As shown in Table 13-14, our method is still highly effective under the domain adaption setting. It is mostly because the VSR improvement caused by domain adaption is mild and therefore cannot fool our verification. Moreover, domain generalization has side effects on the benign accuracy, especially on complicated datasets (*e.g.*, Tiny-ImageNet).

## L    Reproducibility Statement

In the appendix, we provide detailed descriptions of the datasets, models, training and evaluation settings, and computational facilities. We provide the codes and model checkpoints for reproducing the main experiments of our evaluation in the supplementary material. In particular, we also release our training codes at `https://github.com/JunfengGo/Domain-Watermark`.

## M    Societal Impacts

In this paper, we focus on the copyright protection of (open-sourced) datasets. Specifically, we reveal the harmful nature of backdoor-based dataset ownership verification (DOV) and propose the first non-backdoor-based DOV method that is truly harmless. This work has no ethical issues in general since our method is purely defensive and does not reveal any new vulnerabilities of DNNs. However, our method requires a sufficiently large watermarking rate and therefore can not be used to protect a few or a single image. In addition, although our method is resistant to existing adaptive methods, adversaries may try to develop more effective attacks against our DOV method, given the exposure of this paper. People should not be too optimistic about dataset protection.

## N    Discussions about Adopted Data

In this paper, all adopted samples are from the open-sourced datasets (*i.e.*, CIFAR-10, Tiny-ImageNet, and STL-10). The Tiny-ImageNet dataset may contain a few human-related images. We admit that we modified a few samples for watermarking and verification. However, our research treats all samples the same and the verification samples and modified samples have no offensive content. Accordingly, our work fulfills the requirements of these datasets and has no privacy violation.

