# OpenReview forum: "Domain Watermark: Effective and Harmless Dataset Copyright Protection is Closed at Hand"
_NeurIPS.cc/2023/Conference — NeurIPS 2023 poster_

### Official Review · Reviewer_xWCD · 2023-07-05

**Soundness:** 2 fair
**Presentation:** 3 good
**Contribution:** 3 good
**Rating:** 4
**Confidence:** 2

**Summary:**

The authors study copyright projection problem for open datasets. They note that the previous method based on back door attacks are harmful. They propose to watermark some samples such that model training on this sample can correctly classify hard sample, which cannot be correctly classified by benign DNN. Technically, the authors exploit knowledge and techniques in transfer learning to derive domain generator for this task. The authors provide a set theoretical justification and perform a series of experiments to evaluate the proposed scheme.

**Strengths:**

The proposed method is supported by both theory and experiments.

The proposed idea using hard sample for database copyright protection should be novelty.

**Weaknesses:**

The basic idea of this work is illustrated in Fig. 1. (Stage III). If many hard samples (which cannot be classified by benign DNN) can be correctly classified by a suspicious model, it is not necessary that the suspicious model is trained on the copyright dataset. If the owner of suspicious model claims that their model is trained on an extremely large dataset covering numerous hard samples and uses a proprietary architecture, augmentation method and learning scheme, it will be a very reasonable explanation why it can perform better the benign DNN, which is trained on a “small” dataset.  Fig. 3 in fact gives a good illustration. The verification sample of the proposed method in Fig. 3 is clearly a dog to human being and the classification output is a dog. We only can say that the model can handle the hard sample. However, we cannot conclude that the model is trained on the watermarked dataset. Although the paper offers a lot of mathematical justification and experimental results, the authors need to explain the practicality of the proposed method.

Although I do read the paper from beginning to the end, I am not working on this area. I let other reviewers commenting on the technical and experimental issues.

**Questions:**

See the weaknesses.

**Limitations:**

See the weaknesses.

---

> ### Author Rebuttal · Authors · 2023-08-09
>
> Dear Reviewer xWCD, thank you very much for your careful review of our paper and thoughtful comments. We are encouraged by your positive comments on our **novel method**, **theoretical and empirical supports**, and **good contribution**. We hope the following responses can help clarify potential misunderstandings and alleviate your concerns.
>
>
> ---
>
> **Q1**: The basic idea of this work is illustrated in Fig. 1. (Stage III). If many hard samples (which cannot be classified by benign DNN) can be correctly classified by a suspicious model, it is not necessary that the suspicious model is trained on the copyright dataset. If the owner of suspicious model claims that their model is trained on an extremely large dataset covering numerous hard samples and uses a proprietary architecture, augmentation method and learning scheme, it will be a very reasonable explanation why it can perform better the benign DNN, which is trained on a “small” dataset. Fig. 3 in fact gives a good illustration. The verification sample of the proposed method in Fig. 3 is clearly a dog to human being and the classification output is a dog. We only can say that the model can handle the hard sample. However, we cannot conclude that the model is trained on the watermarked dataset. Although the paper offers a lot of mathematical justification and experimental results, the authors need to explain the practicality of the proposed method.
>
> **R1**: Thank you for challenging the foundation of our method! We are deeply sorry that our submission may lead you to some misunderstandings that we hereby want to clarify.
>
> - **Hardly-generalized domain is an exclusive and special domain** where only defenders know what it is. Accordingly, the adversaries (i.e., malicious dataset users) cannot know it and use it to break our verification.
> - Lemma 1 indicates that **we can find a hardly-generalized domain for the dataset, no matter what the model architecture is**. Accordingly, it is not reasonable to claim that a model has good performance on domain-watermarked samples due to its well-designed architecture.
> - As shown in Figures 2-3 in the main manuscript as well as **Figures 2, 5-10 in the Appendix**, **domain-watermarked samples and not natural and very special**. In other words, **it is impossible that dataset users can collect them without using our protected dataset**. Accordingly, it is not reasonable to claim that a model has good performance on domain-watermarked samples because their dataset is extremely large covering numerous hard samples.
> - We admit that using data augmentation or other domain adaption techniques may increase the generalization ability of trained DNNs on domain-watermarked samples, although adversaries have no information of hardly-generalized domains and those augmented samples used for training are significantly different from those of domain-watermarked ones. However, **this improvement is limited since these adapted domains are significantly different from the hardly-generalized domain** (the domain space is high-dimensional and enormous). Accordingly, **this mild improvement cannot break our method**. To further verify it, we evaluate our method on models with classical automatic data augmentations (e.g., color shifting) and domain adaption.
>   - **The Resistance to Data Augmentation**: We adopt strong data augmentations (as shown in the following code block) to train models on benign datasets with the same settings used in our paper. After that, we evaluate their accuracy on our domain-watermarked samples and verify whether our verification method will treat them as malicious in this case. As shown in the following table, **our method will not misjudge (p-value $\gg$ 0.05) although data augmentation can (slightly) increase the accuracy on our domain-watermarked samples**.
>
> ```python!
> #code for data augumentation (written in Torch)
> transforms.RandomHorizontalFlip(), # FLips the image w.r.t horizontal axis
> transforms.RandomRotation(10),#Rotates the image to a specified angel
> transforms.RandomAffine(0, shear=10, scale=(0.8,1.2)), #Performs actions like zooms, change shear angles.
> transforms.ColorJitter(brightness=0.2, contrast=0.2, saturation=0.2), # color shifiting, constrast
> transforms.ToTensor(),
> ```
>
>  | Dataset$\downarrow$, Metric$\rightarrow$                        | BA (%) | Accuracy on DW samples (%) | $\Delta P$ | p-value |
> |:-------------------------------:|:------:|:-------:|:-:|-----------|
> |              CIFAR-10               |     91.50      |     30.15   |   0.76|  1.00      |
> | Tiny-ImageNet |   59.60      |    14.00   |  0.6 |   0.93    |
> | Sub-ImageNet |   75.84      |     6.00  |  0.63|    1.00  |
>
>   - **The Resistance to Domain Adaption/Generalization**: The results show that our method is still highly effective even under domain adption. Please refer to our $R2$ to Reviewer Tjx6 for more details.
> - Domain generalization is still an important yet unsolved problem in computer vision. Accordingly, **there is no existing method can make DNNs generalize to all unseen domains** (with performance on par to that of the source domain) for breaking our method.
>
>
> In conclusion, our method will not misjudge even if users may adopt 'hard samples', data augmentation, and domain adoption. Accordingly, our method is essentially practical. We will add more details and discussions in the appendix of our revision.
>
> ---

---

> > ### Comment · Area_Chair_wxAP · 2023-08-13
> >
> > Dear reviewer xWCD, please take your time to carefully review the author's explanation and provide your response. Thank you!

---

> > ### Comment · Reviewer_xWCD · 2023-08-16
> >
> >
> > "Hardly-generalized domain is an exclusive and special domain where only defenders know what it is. Accordingly, the adversaries (i.e., malicious dataset users) cannot know it and use it to break our verification."
> >
> > How can you guarantee that in this world, no one designs an augmentation method to generate similar hard samples for improving performance?
> >
> >
> > "Lemma 1 indicates that we can find a hardly-generalized domain for the dataset, no matter what the model architecture is. "
> >
> > However, other networks in fact train on other datasets with totally different properties and different size. Is Lemma 1 still appliable?
> >
> >
> > "Figures 2-3 in the main manuscript ....."
> >
> > The verification in Fig. 3 is a dog clearly with special image processing effect (Same for other images in the appendix). However, some stronger networks that can correctly classify it are nothing wrong to me. It just shows that the networks are well trained.
> >
> >
> > "We admit that using data augmentation or other domain adaption techniques may increase the generalization"
> >
> > Using experiments to justify your design may not be enough in this case, because we really do not know how different large and how diverse data on others’ hand. We also have no way to know many different argumentation methods other use.
> >
> >
> > Your approach is somehow like. A professor set some hard questions in exam. If a student can answer most of them, we say that the student is cheating. We assume that there is no genius in the class, no super hard working students, etc. Your answers do not convince me.

---

> > > ### Author Response · Authors · 2023-08-17
> > > **Thank you and further explanations to reviewer's feedback**
> > >
> > > Thank you for your comments and we do understand your concerns. We hereby provide more explanations to further allevate your concerns.
> > >
> > > - We admit that the suspicious model can generalize well on hard samples may because its good training strategies and structures instead of becuase being trained on our protected dataset, since we cannot theoretically verify it. However, **at least the probability of its happening is very small**, as we explained and demonstrated in the rebuttal.
> > >     - **It is unlikely that a benign user can develop data augmentation that can generate (sufficiently) similar hard samples without knowing our specific watermark strategies**, since the space of hardly-generalized domain is huge and our watermark is sample-specific and not unique.
> > >     - Currently, **there is no model that can achieve similar generalizability to humans**, although it was trained on a huge dataset with complicated data augmentation strageties and well-designed structure.
> > >     - Our hypothesis-testing-based verification process ensures that **suspicious models will be treated as trained on our protected dataset only when they have sufficiently high (not just certain) generalizability over a large number (not just a small number) of hard samples**. Accordingly, our method can reduce the side effects of randomness to a large extent.
> > > - **It does not diminish the practicality of our method, even if this rare event is likely to happen**. In our opinion, using verification-based defenses (in a legal system) requires an official institute for arbitration. Specifically, all commercial models should be registered here, based on the unique identification (e.g., MD5 code) of their model files and training datasets, before being used online. When this official institute is established, its staff should take responsibility for the verification process. For example, they can require the company to provide the dataset with the same registered identification and then check whether it contains our protected samples (via algorithms). Of course, if the suspect model is proven to be benign, the user will need to pay a settlement to its company to prevent casual malicious ownership verification. In this case, even if our method misclassifies in rare cases, it does not compromise the interest of the suspect model. **Our method is practical in this realistic situation, as long as it has a sufficiently high probability of correct verification**.

---

> > > > ### Comment · Reviewer_xWCD · 2023-08-19
> > > >
> > > > Thank you very much for your response and admit “that the suspicious model can generalize well on hard samples may because its good training strategies and structures instead of because being trained on our protected dataset, since we cannot theoretically verify it.” Although I increase one mark, I cannot fully verify your statements. As I mentioned in the original review, I am not working in this topic and another two reviewers also have low confidence. I recommend AC seeking some opinions from other reviewers or researchers, who are working this topic or highly related areas for this issue.

---

> > > > > ### Author Response · Authors · 2023-08-19
> > > > >
> > > > > Thank you for your quick responses and previous valuable comments! We are glad that our responses at least partly alleviate your concerns. We also sincerely thank you for explaining that your relative low score and confidence are because you cannot fully verify our statements and understand our paper. We fully understand it because dataset ownership verification is really a very new research area. Thank you again for trying very hard to review our work and help us improving it!

---

> ### Author Response · Authors · 2023-08-14
> **Thanks to Reviewer xWCD**
>
> Please allow us to thank you again for reviewing our paper and the valuable feedback, and in particular for recognizing the strengths of our paper in terms of novel method, theoretical and empirical supports, and good contribution.
>
> Kindly let us know if our response and the new experiments have properly addressed your concerns. We are more than happy to answer any additional questions during the post-rebuttal period. Your feedback will be greatly appreciated.

---

### Official Review · Reviewer_FQjV · 2023-07-05

**Soundness:** 3 good
**Presentation:** 2 fair
**Contribution:** 3 good
**Rating:** 5
**Confidence:** 4

**Summary:**

This paper first revisits the backdoor-based dataset ownership verifications and claims that these methods are harmful since they introduce the malicious misclassification. Given this, authors propose a truly harmless dataset copyright protection via making watermarked models correctly classify hard samples which will be misclassified by the benign models. They first generate domain watermark and then modify a set of benign samples which have similar effects to domain watermarked samples to insert the watermark into the dataset. Finally, they propose a hypothesis testing to verify whether a suspect model has unauthorized dataset usage. Experiments on CIFAR-10, Tiny-ImageNet and STL-10 show the effectiveness.

**Strengths:**

1. The proposed method is novel to some extend. This paper proposes to protect the intellectual property of dataset from another perspective, i.e., making watermarked models correctly classify hard samples which will be misclassified by the benign models, differed from previous backdoor-based methods.
2. The theoretical analysis is reasonable and the experiments are extensive.

**Weaknesses:**

Section 3.3 and 3.4 are not clear and well-written.  For example, (a) lemma 1 (line 172~176) mentions the z and \hat_{z}, the features for source and target domains. However, the specific meaning of z/\hat_{z} and how to obtain z/\hat_{z} are not explained until line 189; (b) it's better to give the specific meanings of \theta, \omega and f in equation 2;

Although authors have conducted extensive experiments, it's better to test with more model structures instead of only two (VGG and ResNet). Besides, Section 5.3.2 in the main manuscript and Section J in the appendix discuss the resistance to some potential adaptive methods. However, I think the true adaptive method are out-of-domain detection in the inference stage which does not involved in either main manuscript or the appendix.

**Questions:**

1. In Equation (2), it seems that both z and \hat_{z} are relative to \omega while in the second term of \omega * , i.e., , \hat_{z} is not relative to \omega.
2. In line 262, authors set the watermarking rate \gamma=0.1 for their method and other baselines. However, for the most of baselines, they need to set one class as the target class, which means all of samples in the target class are perturbed for 10-class datasets (CIFAR-10 and STL-10) and a few classes are selected to be target classes for Tiny-ImageNet (200-class). I wonder the concrete settings.
3. In the Experiments, authors describe that they conduct experiments with two model structures VGG and ResNet, while results in table 1&2 do not reflect this setting, and the detailed settings of these results are not clear.

**Limitations:**

Please refer to the questions above.

---

> ### Author Rebuttal · Authors · 2023-08-09
>
>
> Dear Reviewer FQjV, thank you very much for your careful review of our paper and thoughtful comments. We are encouraged by your positive comments on our **novel method**, **reasonable theoretical analysis**, **extensive experiments**, **good soundness**, and **good contribution**. We hope the following responses can help clarify potential misunderstandings and alleviate your concerns.
>
> ---
> **Q1**: Provide more details in Section 3.3-3.4.
>
> **R1**: Thank you for you constructive suggestions! We will provide a more clear definition of $z$ and $\hat{z}$ and how to obtain them in Lemma 1. Besides, we will provde the definition of $\theta$ (i.e., parameters of the domain generator), $\omega$ (i.e., parameters of the model $f$), and $f$ in Eq.(2).
>
> ---
>
> **Q2**: Although authors have conducted extensive experiments, it's better to test with more model structures instead of only two.
>
> **R2**: Thank you for this constructive suggestion! We evaluate our method with two additional representative model structures (i.e., MobileNet-V2 and DenseNet-40) following our default settings. As shown in the following tables, **our method is still highly effective with them**. We will provide more details in the appendix of our revision.
>
> **Table 1.** The watermark performance of our domain watermark with MobileNet-V2 and DenseNet-40 on CIFAR-10.
> |                          | BA (%) | VSR (%) | $H$ | $\hat{H}$ |
> |:-------------------------------:|:------:|:-------:|:-:|-----------|
> |                MobileNet-V2             |   88.29     |     82.5    |  0.18 | -0.58          |
> |  DenseNet-40 |    90.04        |      87.10   |0.13   |    -0.70       |
>
>
> **Table 2.** The verification effectiveness via our domain watermark with MobileNet-V2 and DenseNet-40 on CIFAR-10.
> |           Structure$\rightarrow$          |   MobileNet   |   MobileNet   |  MobileNet |    DenseNet   |    DenseNet   |  DenseNet  |
> |:-----------------------------------------:|:-------------:|:-------------:|:----------:|:-------------:|:-------------:|:----------:|
> | Metric$\downarrow$, Scenario$\rightarrow$ | Independent-D | Independent-M |  Malicious | Independent-D | Independent-M |  Malicious |
> |                 $\Delta P$                |      0.74     |      0.74     |    0.09    |      0.72     |      0.74     |    0.08    |
> |                  p-value                  |       1       |       1       | $10^{-19}$ |       1       |       1       | $10^{-19}$ |
>
> ---
>
> **Q3**: I think the true adaptive method are out-of-domain detection in the inference stage which does not involved in either main manuscript or the appendix.
>
> **R3**: Thank you for your suggestion and we do understand your concerns.
> - We admit that using out-of-domain generators may detect our domain-watermarked samples. However, since only defenders know the detailed domain for watermarked samples, **the adversaries can only train a out-of-domain generator that would detect and reject samples from all domains other than the source domain**. We argue **it is rare to apply such a generator in practice**, particularly in computer vision. This is because domain generalization capability plays a crucial role in real-world DNN applications, **failing to give accurate predictions on unseen domains' samples would significantly reduce its real-world performance**.
> - Moreover, we do understand your concerns. **We have also considered the adaptvie attack and showed the resistance of our method to it** during the rebuttal. Please see the $R1$ to Reviewer Tjx6 for more details.
>
>
> ---
>
> **Q4**: In Equation (2), it seems that both $z$ and $\hat{z}$ are relative to $\omega$ while in the second term of $\omega^*$ , i.e., , $\hat{z}$ is not relative to $\omega$.
>
> **R4**: Thank you for pointing it out! We will repalce $\hat{z}$ with $\hat{z}(\omega)$ to avoid potential misunderstanding.
>
> ---
>
> **Q5**: In line 262, authors set the watermarking rate $\gamma=0.1$ for their method and other baselines. However, for the most of baselines, they need to set one class as the target class, which means all of samples in the target class are perturbed for 10-class datasets (CIFAR-10 and STL-10) and a few classes are selected to be target classes for Tiny-ImageNet (200-class). I wonder the concrete settings.
>
> **R5**: We are deeply sorry that our submission may lead you to some misunderstandings that we want to clarify here.
>
> - **Only the label-consistent attack (instead of most methods) generates watermark samples solely based on benign samples from the target class**. Specifically, poisoned-label watermarks (i.e., BadNets, Blended, WaNet, and UBW-P) can add trigger patterns to samples from all classes; Although they are clean-label watermarks, Sleeper Agent, UBW-C, and our DW can generate watermarked samples from all classes since we formulate watermark generation as a bi-level optimization problem. We will provide more details in the appendix of our revision to avoid potential misunderstandings.
> - We have to admit that we forgot to include a setting detail that **we poison 50\% samples from the target class (instead of 10\% samples from the whole dataset) for the label-consistent attack on all datasets**, following the default settings used in 'BackdoorBox'. We will add this detail in Section 5.1 in our revision.
>
> ---
>
> **Q6**: In the Experiments, authors describe that they conduct experiments with two model structures VGG and ResNet, while results in table 1&2 do not reflect this setting, and the detailed settings of these results are not clear.
>
> **R6**: We are deeply sorry that our submission may lead you to some misunderstandings. In fact, **we forget to include a 'respectively' at the end of the first sentence in Section 5**. In other words, we conduct experiments on two benchmark datasets, including CIFAR-10 and Tiny-ImageNet with VGG and ResNet, respectively. We will correct it in our revision to avoid potential misunderstandings.
>
> ---

---

> > ### Author Response · Authors · 2023-08-20
> >
> > Dear Reviewer FQjV,
> >
> > Thank you once again for your valuable time and constructive comments. We would like to kindly inform you that we should have already addressed your concerns in our rebuttal.
> >
> > As the reviewer-author discussion phase is nearing to the end, we would like to know whether our explanations and experiments have properly addressed your concerns. We are more than happy to answer any additional questions during the post-rebuttal period. Your feedback will be greatly appreciated.

---

> ### Author Response · Authors · 2023-08-14
> **Thanks to Reviewer FQjV**
>
> Please allow us to thank you again for reviewing our paper and the valuable feedback, and in particular for recognizing the strengths of our paper in terms of novel method, reasonable theoretical analysis, extensive experiments, good soundness, and good contribution.
>
> Kindly let us know if our response and the new experiments have properly addressed your concerns. We are more than happy to answer any additional questions during the post-rebuttal period. Your feedback will be greatly appreciated.

---

### Official Review · Reviewer_Tjx6 · 2023-07-06

**Soundness:** 3 good
**Presentation:** 3 good
**Contribution:** 3 good
**Rating:** 5
**Confidence:** 2

**Summary:**

This paper presents a method to use invisible domain watermarks for dataset ownership verification. Compared to previous methods, the motivation of this paper is to avoid harmful practice which negatively affects the model performance. The method is validated on CIFAR and Tiny-Imagenet datasets.

**Strengths:**

1.[reference] The paper provides a comprehensive literature review.

2.[clarity] Extensive experimental results and ablation studies demonstrate the effectiveness of the proposed method.

3.[clarity] The paper has a clear motivation, which is to avoid harmful practice compared to existing methods. In particular, the proposed method is invisible, clear-label, and effective.

**Weaknesses:**


1.[technical soundness] Adaptive attack against this proposed method is not discussed.

2.[limitation] The paper relies on assumption on generalization. For standard training the assumptions should hold. But what if the dataset to protect is used for domain adaptation, which has higher generalization capability across domains? This is not adaptive attack by design, but it might coincidently break the core assumptions of the proposed method.

**Questions:**

Overall, the paper is comprehensive.
My main concerns are about the adaptive attacks and domain generalization training methods coincidently breaking the assumption of the proposed method.

**Limitations:**

See weaknesses.

---

> ### Author Rebuttal · Authors · 2023-08-09
>
> Dear Reviewer Tjx6, thank you very much for your careful review of our paper and thoughtful comments. We are encouraged by your positive comments on our **comprehensive review**, **extensive experiments**, **effective method**, **good presentation**, **good soundness**, and **good contribution**. We hope the following responses can help clarify potential misunderstandings and alleviate your concerns.
>
> ---
>
> **Q1**: Adaptive attack against this proposed method is not discussed.
>
> **R1**: Thank you for your comments and we do understand your concerns.
> - In general, our method is relatively complicated with two bi-level optimizations. Accordingly, **it is very difficult or even impossible to design a penalty term corresponding to the process of our method to design adaptive attacks**. It is why we only evaluate the resistance of our method to five representative existing defenses that could be effective in defending against our dataset watermark.
> - From a high-level perspective, our method intends to make DNNs generalizing to a specific domain (i.e., hardly-generalized domain) with samples from the source domain under the standard training process (since similar-effect samples are similar to clean ones). Accordingly, we can design adaptive attack by supressing domain generalization during the training process. We hereby exploit [non-transferable learning](https://arxiv.org/pdf/2106.06916.pdf) for the design. As shown in following tables, **our method is still highly effective under non-transferable learning**.
>
>
> | Task$\downarrow$, Metric$\rightarrow$                        | BA (%) | VSR (%) | $H$ | $\hat{H}$ |
> |:-------------------------------:|:------:|:-------:|:-:|-----------|
> |                CIFAR-10               |  90.01      |     86.40    | 0.14 | -0.67          |
>
> | Metric$\downarrow$, Scenario$\rightarrow$ | Independent-D | Independent-M | Malicious  |
> |--------------------|-------------------------------------------|---------------|---------------|
> | $\Delta P$                                          |     0.86    | 0.80         |       0.08
> | p-value                                          | 1             | 1            | $10^{-21}$ |
>
> We will provide more details and discussions in the appendix of our revision.
>
>
>
>
>
> ---
>
> **Q2**: The paper relies on assumption on generalization. For standard training the assumptions should hold. But what if the dataset to protect is used for domain adaptation, which has higher generalization capability across domains? This is not adaptive attack by design, but it might coincidently break the core assumptions of the proposed method.
>
> **R2**: Thank you for this insightful question! We here adopt [L2D](https://arxiv.org/abs/2108.11726) as the representative method to investigate the robustness of our domain watermark against domain generalization. Specifically, we train both the watermarked and benign DNNs with L2D. The results are as follows.
>
> |  Task$\downarrow$, Metric$\rightarrow$                        | BA (%) | VSR (%) | Accuracy on other hardly generalized domain samples (%) |$H$ | $\hat{H}$ |
> |:-------------------------------:|:------:|:------:|:-------:|:-:|-----------|
> |                CIFAR-10                  |     90.10    |  86.10 | 39.10 |-0.14     | -0.73   |
> |  Tiny-ImageNet |    48.63        |   42.00  |16.00| 0.58     |  -0.26       |
>
>
> | Metric$\downarrow$ | Task$\downarrow$, Scenario$\rightarrow$ | Independent-D | Independent-M | Malicious  |
> |--------------------|-------------------------------------------|---------------|---------------|------------|
> | $\Delta P$         | CIFAR-10                                 |     0.63    | 0.60         |       0.05
> | $\Delta P$         | Tiny-ImageNet                                  | 0.53          | 0.53         | 0.07      |
> | p-value            | CIFAR-10                                   | 0.90            | 0.71            | $10^{-33}$ |
> | p-value            | Tiny-ImageNet                                  | 0.96              | 0.95            | $10^{-12}$ |
>
>
> The above results indiate that **our method is still highly effective under the domain adaption setting**. It is mostly because the VSR imporvement caused by domain adaption is mild and therefore cannot fool our verification. Moreover, **domain generalization has side effects on the benign accuracy**, especially on complicated datasets ($e.g.,$ Tiny-ImageNet).
>
> ---

---

> > ### Author Response · Authors · 2023-08-20
> >
> > Dear Reviewer Tjx6,
> >
> > Thank you once again for your valuable time and constructive comments. We would like to kindly inform you that we should have already addressed your concerns in our rebuttal.
> >
> > As the reviewer-author discussion phase is nearing to the end, we would like to know whether our explanations and experiments have properly addressed your concerns. We are more than happy to answer any additional questions during the post-rebuttal period. Your feedback will be greatly appreciated.

---

> ### Author Response · Authors · 2023-08-14
> **Thanks to Reviewer Tjx6**
>
> Please allow us to thank you again for reviewing our paper and the valuable feedback, and in particular for recognizing the strengths of our paper in terms of comprehensive review, extensive experiments, effective method, good presentation, good soundness, and good contribution.
>
> Kindly let us know if our response and the new experiments have properly addressed your concerns. We are more than happy to answer any additional questions during the post-rebuttal period. Your feedback will be greatly appreciated.

---

### Official Review · Reviewer_DDiA · 2023-07-25

**Soundness:** 2 fair
**Presentation:** 3 good
**Contribution:** 2 fair
**Rating:** 6
**Confidence:** 2

**Summary:**

This paper proposes domain watermark (DW), a new watermark technique for dataset verification. Domain watermark focuses on watermarking the dataset without introducing the harmful side effect. This paper first proposes to generate the hardly generalized domain by minimizing the mutual information between the dataset and hardly generalized domain. Then this paper perturbs some of the inputs to have similar effects as the domain-watermarked samples. Then the dataset contains its original domain and the watermarked domain. The model trained on it also learns the watermarked domain can be exposed by correctly classifying the watermarked domain images.
The evaluation is done on CIFAR-10 and Tiny-ImageNet and shows that the watermarked model can be effectively distinguished from model trained on data from another dataset or model trained on the unwatermarked dataset.

**Strengths:**

I think the area this paper studies is important and I agree that the watermark should not introduce harms or vulnerabilities that can be exploited by the attackers.

In general, I like the idea of embedding the model with a harmless distribution that is orthogonal to the main distribution as the watermark.

This paper is well-written and easy to understand.

**Weaknesses:**

1. My biggest concern with this approach is that it just optimizes to find the most hardly generalized domain. It is possible that two watermark domains on two datasets of one similar domain collide.
Suppose there are two dataset owners A and B. They have dataset from the similar domains and they both use this paper’s method to do the domain watermark. An attacker C trains on A’s dataset, would B’s WM images also have high VSR on C’s model?
If so, it would be hard for A to claim that C trained on A’s model as the high VSR on the WM images does not prove the model trained on A (it could also be trained on B or trained on another model from the similar domain).
I suggest adding a new experiment on this hypothetical scenario or include a new mechanism to avoid/reduce the possibility of collision of the watermark domains.

2. The general idea of this paper is that the watermarked dataset contains two distributions, the dataset’s domain and the watermark domain. Many of the existing backdoor poison samples detection methods also depend on this assumption. Will the watermark domain bypass some of the existing backdoor detection techniques, e.g.[1-3]? I suggest this paper to be evaluated against backdoor samples detection techniques.

3. I am not able to find the clear definition of the Harmful degree and relative harmful degree. Could you put these definitions in the main text?


[1] Effective Backdoor Defense by Exploiting Sensitivity of Poisoned Samples
[2] Demon in the variant: Statistical analysis of {DNNs} for robust backdoor contamination detection
[3] Pre-activation Distributions Expose Backdoor Neurons


**Questions:**

Please see the questions I raise in the weakness section.


**Limitations:**

This paper discusses the limitation of this paper is bounded by the accuracy of the watermark models.

---

> ### Author Rebuttal · Authors · 2023-08-09
>
> Dear Reviewer DDiA, thank you very much for your careful review of our paper and thoughtful comments. We are encouraged by your positive comments on our **significant topic**, **interesting idea**, and **clear writing**. We hope the following responses can help clarify potential misunderstandings and alleviate your concerns.
>
>
>
> ---
> **Q1**: My biggest concern with this approach is that it just optimizes to find the most hardly generalized domain. It is possible that two watermark domains on two datasets of one similar domain collide. Suppose there are two dataset owners A and B. They have dataset from the similar domains and they both use this paper’s method to do the domain watermark. An attacker C trains on A’s dataset, would B’s WM images also have high VSR on C’s model? If so, it would be hard for A to claim that C trained on A’s model as the high VSR on the WM images does not prove the model trained on A (it could also be trained on B or trained on another model from the similar domain). I suggest adding a new experiment on this hypothetical scenario or include a new mechanism to avoid/reduce the possibility of collision of the watermark domains.
>
>
> **R1**: Thank you for this insightful question! We are deeply sorry that our submission may lead you to some misunderstandings that we want to clarify here.
>
> 1. We intend to obtain a hardly-generalized domain instead of the most hardly-generalized one for the given dataset.
> 2. **Hardly-generalized domain is not unique** for each given dataset. As shown in our Appendix C, we can easily generate many different hardly-generalized domains with different random seeds. Accordingly, **it is almost impossible for two independent defenders to find the same (or even similar) hardly-generalized domain even for the same unprotected original dataset**.
> 3. In particular, we have considered this potential problem in our method. **The second term in our Eq.(7) is to prevent the watermarked model can achieve a similar generalization performance on (other) unseen domains** as the target domain to  preserve its uniqueness for verification purposes.
> 4. In our Table 2 (Independent-D), **we have verified that we cannot fool our verification method with watermarked samples from another hardly-generalized domain (crafted from the same benign dataset) that is different from the one used for defender**. Besides, we have also conducted more experiments in Appendix I.2 to further verify it.
>
>
> ---
> **Q2**: The general idea of this paper is that the watermarked dataset contains two distributions, the dataset’s domain and the watermark domain. Many of the existing backdoor poison samples detection methods also depend on this assumption. Will the watermark domain bypass some of the existing backdoor detection techniques, e.g.[1-3]? I suggest this paper to be evaluated against backdoor samples detection techniques.
>
> **R2**: Thank you for this insightful question! We hereby provide more explanations to alleviate your concerns.
> - In our paper (Section 5.3.2 and Appendix J), we have evaluated our method against five representative backdoor defenses, including three detection methods (i.e., ShrinkPad, Scale-UP, and Neural Cleanse). The results showed that **our method is resistant to all of them**.
> - To further alleviate your concerns, we evaluate whether our method is resistant to your suggested defenses [2-3]. Since the rebuttal time is limited, we don't test D-BR [1] and D-ST [1] due to their complexity of codes and high computational consumption (due to its self-supervised stage).
>   - **The Resistance to BNP [2]**：We implement BNP based on its open-sourced codes with its default settings on CIFAR-10. With BNP, the benign accuracy drops from 91.06\% to 85.86\% while the verification success rate (VSR) drops from 90.4\% to 80.1\%, resulting in the p-value of identifying dataset stealing as $10^{-18}$ ($\ll 0.01$). **These results show that our method is resistant to BNP**. It is probably because our method doesn't introduce adverse features to the target DNNs as the verified samples share most semantic features with their benign versions according to Eq.(2). As such, the 'bad neurons' caused by our approach closely collided with the 'benign neurons'.
>   - **The Resistance to SCAn [3]**：We implement SCAn also on CIFAR-10 with different budgets for clean samples required by SCAn. As shown in the following table, **SCAn can not identify the watermarked DNN ($Ln(J^{*})\leq 2)$ and therefore failing to further filter out watermarked samples**, even with 20\% of clean samples for each class. It is probably because our modified samples are clean-label and invisible and their features contain no adverse information.
>
>
> |   Budgets for Clean Samples    |  1%  |   5%  |  10% |  15% |  20% |
> |:-----:|:----:|:-----:|:----:|:----:|:----:|
> |$Ln(J^{*})$ | 1.21 | 1.209 | 1.27 | 1.49 | 1.51 |
>
>
> We will add more details and discussions in the appendix of our revision.
>
>
> ---
>
> **Q3**: I am not able to find the clear definition of the Harmful degree and relative harmful degree. Could you put these definitions in the main text?
>
> **R3**: We have provided their definitions in Definition 1 from Section 3.2 (Line 149-154, Page 4). You might have accidentally missed it :)
>
> ---

---

> > ### Author Response · Authors · 2023-08-20
> >
> > Dear Reviewer DDiA,
> >
> > Thank you once again for your valuable time and constructive comments. We would like to kindly inform you that we should have already addressed your concerns in our rebuttal.
> >
> > As the reviewer-author discussion phase is nearing to the end, we would like to know whether our explanations and experiments have properly addressed your concerns. We are more than happy to answer any additional questions during the post-rebuttal period. Your feedback will be greatly appreciated.

---

> ### Author Response · Authors · 2023-08-14
> **Thanks to Reviewer DDiA**
>
> Please allow us to thank you again for reviewing our paper and the valuable feedback, and in particular for recognizing the strengths of our paper in terms of significant topic, interesting idea, and clear writing.
>
> Kindly let us know if our response and the new experiments have properly addressed your concerns. We are more than happy to answer any additional questions during the post-rebuttal period. Your feedback will be greatly appreciated.

---

### Official Review · Reviewer_a3iY · 2023-07-25

**Soundness:** 3 good
**Presentation:** 3 good
**Contribution:** 3 good
**Rating:** 6
**Confidence:** 2

**Summary:**

The authors propose a novel dataset ownership verification method that relies on correctly classifying 'hard' samples rather than classifying 'easy' samples which is harmless to the benign data. The proposed method is inspired by the generalization property of DNNs.  Experiments verify the effectiveness of proposed method.

**Strengths:**

1)	Interesting insight. The insight of the harmless DOV method is interesting.
2)	Clear writing: The paper is overall well-organized and easy to follow. The authors discuss the details of their design choices.


**Weaknesses:**

1. While hyper-parameter selection in ablation studies section is crucial, I think the impact of the module block on the algorithm more intriguing. Specifically, I am interested in understanding how the generation of similar-effect samples during the creation of the protected dataset affects the final performance.

2. I think that the Domain Generator plays a crucial role in the overall algorithm. Therefore, it is essential to address the performance evaluation of this module.

3. The experiments were conducted on relatively small datasets (e.g., CIFAR-10 and Tiny-ImageNet). The transferability of the algorithm's effectiveness to larger datasets remains a question. For instance, it is crucial to examine whether the training of the Domain Generator is affected when applied to larger datasets. I believe this issues should be addressed at least in the "Discussion" section.



**Questions:**

1)  Similar-effect samples' influence to the final performance.
2) The performance of the proposed method on the larger dataset.

**Limitations:**

See part ‘Strengths And Weaknesses’.

---

> ### Author Rebuttal · Authors · 2023-08-09
>
> Dear Reviewer a3iY, thank you very much for your careful review of our paper and thoughtful comments. We are encouraged by your positive comments on our **interesting insight**, **clear writing**, **good soundness**, and **good contribution**. We hope the following responses can help clarify potential misunderstandings and alleviate your concerns.
>
>
> ---
> **Q1**: While hyper-parameter selection in ablation studies section is crucial, I think the impact of the module block on the algorithm more intriguing. Specifically, I am interested in understanding how the generation of similar-effect samples during the creation of the protected dataset affects the final performance.
>
> **R1**: Thank you for this insightful comment! We agree that understanding the impact of modules is also important. We hereby provide more details and dicussions.
>
> - In general, **using similar-effect samples will decrease the effectiveness of our method**. Specifically, the most direct and effective way for learning the hardly-generalized domain is to use domain-watermarked samples. The better the learning of this domain, the better the verification effectiveness. However, the approximation cannot be perfect, although we design an effective bi-level optimization to approximate the effects of domain-watermarked samples with similar-effect samples. To further verify it, we compare the performance of our domain watermark (DW) and that of its variant using domain-watermarked samples directly. As shown in the following table and Table 1 in the rebuttal PDF, **the performance of DW is (slightly) lower than that of its variant**.
>
> **Table 1.** The watermark performance of DW with (w/) and without (w/o) using similar-effect samples on CIFAR-10.
> |  Method$\downarrow$, Metric$\rightarrow$                        | BA (%) | VSR (%) | $H$ | $\hat{H}$ |
> |:-------------------------------:|:------:|:-------:|:-:|-----------|
> |                DW (w/)               |   90.86     |     90.45    |  0.10 | -0.77          |
> |  DW (w/o) |    **91.61**        |     **91.40**   | **0.09**   |   **-0.78**       |
>
>
>
> - **Generating similar-effect samples instead of directly using domain-watermarked samples is necessary for ensuring watermark stealthiness**. As shown in Figure 2-3 in our main manuscript, the domain-watermarked image is visually very different from its original version, whereas similar-effect ones are close to their original version.
>
> - To further evaluate the similar-effect generation module, we hereby report the **optimizaiotn loss** (i.e., gradient matching loss) (between domain-watermarked samples and their similar-effect versions) and the **verification success rate** (for domain-watermarked samples) with respect to the optimization procedure of Eq.(7). As shown in Figure 1 of the rebuttal PDF, our generation module can generate similar-effect (i.e., low gradient matching loss) invisible samples while achieveing sufficiently high verificaiton success rate for their domain-watermarked versions.
>
> We will provide more details and discussions in the appendix of our revision.
>
>
> ---
>
>
> **Q2**: I think that the Domain Generator plays a crucial role in the overall algorithm. Therefore, it is essential to address the performance evaluation of this module.
>
>
> **R2**: Thank you for this insightful comment! We agree that understanding the impact of domain generator is also important. We hereby provide more details and dicussions.
>
> - Different from generating similar-effect samples, **domain generator is indispensable to our method**. Accordingly, we cannot evaluate its performance by testing the performance of our method without this module.
> - To further evaluate this module, we hereby report the **mutual information** (between benign samples and their domain-watermarked version) and the **accuracy** (of generated domain-watermarked samples) with respect to the optimization procedure of Eq.(2). We adopt CIFAR-10 and ImageNet to evaluate this module with VGG-16 and ResNet-34, respectively. As shown in Figure 2 of the rebuttal PDF, our module can craft hardly-generalized domain-watermarked samples with a low mutual information (and therefore a low classification accuracy) for benign models.
>
>
> We will add more details and discussions in the appendix of our revision.
>
>
> ---
> **Q3**: The experiments were conducted on relatively small datasets (e.g., CIFAR-10 and Tiny-ImageNet). The transferability of the algorithm's effectiveness to larger datasets remains a question. For instance, it is crucial to examine whether the training of the Domain Generator is affected when applied to larger datasets. I believe this issues should be addressed at least in the "Discussion" section.
>
> **R3**: Thank you for this insightful comment! To further alleviate your concerns, we hereby evaluate our method on Sub-ImageNet. It contains 50 random classes of samples within ImageNet whose image size is $224 \times 224$. As shown in the following table and Table 2 in rebuttal PDF, **our method is still highly effective on Sub-ImageNet** following our default settings. We also report the performacne of the domain generator module on Sub-ImageNet in Figure 2 in the Rebuttal PDF. These results indicate that our method can be effective on large datasets.
>
> **Table 2.** The watermark performance of our domain watermark on Sub-ImageNet.
> |  Model$\downarrow$, Metric$\rightarrow$                        | BA (%) | VSR (%) | $H$ | $\hat{H}$ |
> |:-------------------------------:|:------:|:-------:|:-:|-----------|
> |                ResNet-18               |  78.64      |     72.00    | 0.28 | -0.68          |
> |  ResNet-34 |   75.56      |     68.00  | 0.32  |  -0.64       |
>
>
>
>
> We will add more details and discussions in the appendix of our revision.
>
>
> ---

---

> > ### Author Response · Authors · 2023-08-20
> >
> > Dear Reviewer a3iY,
> >
> > Thank you once again for your valuable time and constructive comments. We would like to kindly inform you that we should have already addressed your concerns in our rebuttal.
> >
> > As the reviewer-author discussion phase is nearing to the end, we would like to know whether our explanations and experiments have properly addressed your concerns. We are more than happy to answer any additional questions during the post-rebuttal period. Your feedback will be greatly appreciated.

---

> ### Author Response · Authors · 2023-08-14
> **Thanks to Reviewer a3iY**
>
> Please allow us to thank you again for reviewing our paper and the valuable feedback, and in particular for recognizing the strengths of our paper in terms of interesting insight, clear writing, good soundness, and good contribution.
>
> Kindly let us know if our response and the new experiments have properly addressed your concerns. We are more than happy to answer any additional questions during the post-rebuttal period. Your feedback will be greatly appreciated.

---

> ### Author Response · Authors · 2023-08-17
> **A Gentle Reminder of the Final Feedback**
>
> Thank you very much again for your initial comments. They are extremely valuable for improving our work. We shall be grateful if you can have a look at our response and modifications, and kindly let us know if anything else that can be added to our next version.

---

### Author Rebuttal · Authors · 2023-08-09

Two figures and tables are contained in the rebuttal PDF.

---

### Decision · Program_Chairs · 2023-09-21

**Decision:**

Accept (poster)

**Comment:**

Overall, the reviewers agree that this submission studies an important and very interesting research problem, proposes a novel method, has a high writing quality, conducts sufficient experiments and literature review, and addresses most of the reviewers' concerns well through rebuttal. One reviewer raised a concern about the possible ineffectiveness of the authors' approach in some extreme cases, which the authors also responded convincingly.